# The protein architecture of the endocytic coat analyzed by FRET microscopy

Michal Skruzny[1,2,*] iD, Emma Pohl[1,2], Sandina Gnoth[1,2], Gabriele Malengo[1,2] & Victor Sourjik[1,2]

## Abstract

Endocytosis is a fundamental cellular trafficking pathway, which requires an organized assembly of the multiprotein endocytic coat to pull the plasma membrane into the cell. Although the protein composition of the endocytic coat is known, its functional architecture is not well understood. Here, we determine the nanoscale organization of the endocytic coat by FRET microscopy in yeast *Saccharomyces cerevisiae*. We assessed pairwise proximities of 18 conserved coat-associated proteins and used clathrin subunits and protein truncations as molecular rulers to obtain a high-resolution protein map of the coat. Furthermore, we followed rearrangements of coat proteins during membrane invagination and their binding dynamics at the endocytic site. We show that the endocytic coat proteins are not confined inside the clathrin lattice, but form distinct functional layers above and below the lattice. Importantly, key endocytic proteins transverse the clathrin lattice deeply into the cytoplasm connecting thus the membrane and cytoplasmic parts of the coat. We propose that this design enables an efficient and regulated function of the endocytic coat during endocytic vesicle formation.

**Keywords** clathrin; endocytosis; FRET; membrane reshaping; yeast
**Subject Categories** Membranes & Trafficking; Organelles
**Mol Syst Biol. (2020) e9009**

## Introduction

The knowledge of the nanoscale organization of cellular machines is crucial for understanding their function and mechanism of action. This certainly applies to the endocytic machinery, which is responsible for molecular uptake, signaling, and membrane homeostasis in all eukaryotic cells (McMahon & Boucrot, 2011; Mettlen *et al*, 2018). The endocytic machinery consists of dozens of proteins, which assemble beneath the plasma membrane, at the endocytic site, to mediate the endocytic site initiation, cargo selection, membrane invagination, and finally vesicle scission and uncoating (Kaksonen & Roux, 2018).

The functionally paramount protein assembly at the endocytic site is the endocytic coat complex made of the clathrin lattice and multiple endocytic adaptor and scaffold proteins. This endocytic coat is essential for setting up the endocytic site, for cargo selection, and for membrane invagination (Kirchhausen *et al*, 2014; Merrifield & Kaksonen, 2014; Robinson, 2015). Initially, the clathrin lattice was thought to be the main structural element of the coat, assisted by adaptor proteins that connect it to the membrane. However, observations of rapid rearrangements of the clathrin lattice at the endocytic site have challenged this view (Wu *et al*, 2001; Avinoam *et al*, 2015; Scott *et al*, 2018). Moreover, endocytic adaptors were shown to form regular assemblies on the plasma membrane independently of clathrin (Skruzny *et al*, 2015; Garcia-Alai *et al*, 2018). Thus, the functional organization of the endocytic coat remains to be understood.

The analysis of the protein architecture of the endocytic coat is complicated by its complex, compact, and dynamic character. Even the simple endocytic coat of budding yeast *Saccharomyces cerevisiae* contains more than 20 proteins (each in dozens of copies) that are densely packed in a hemisphere with the radius of 30–50 nm and associated together for a limited time of 20–40 s (Kaksonen *et al*, 2005; Idrissi *et al*, 2008, 2012; Boettner *et al*, 2012; Kukulski *et al*, 2012; Weinberg & Drubin, 2012; Picco *et al*, 2015; Sun *et al*, 2019). Hence, while recent super-resolution and electron microscopy studies brought great progress in the understanding the lateral organization of the endocytic site (Sochacki *et al*, 2017; Mund *et al*, 2018), these techniques still did not possess enough resolution to describe the molecular architecture of the centrally located endocytic coat.

Importantly, densities and ratios of coat proteins at the endocytic site are well suited for mapping their organization by Förster resonance energy transfer (FRET), a non-radiative energy transfer from an excited donor fluorophore to a nearby acceptor fluorophore. FRET specifically occurs between fluorophores separated by < 10 nm and therefore allows the detection of nanometer distances between fluorescently tagged molecules. FRET has already been successfully applied to systematically map the *in vivo* organization of several protein machines, such as kinetochore, spindle pole body, contractile ring, and chemotactic sensory system (Muller *et al*, 2005; Kentner & Sourjik, 2009; Aravamudhan *et al*, 2014; Gryaznova *et al*, 2016; McDonald *et al*, 2017).

Here, we applied *in vivo* FRET microscopy to determine the nanoscale architecture of the multiprotein endocytic coat. We analyzed 227 pairwise proximities between the N- and/or C-termini of 18 conserved coat-associated proteins at the yeast endocytic site,

---

1  Department of Systems and Synthetic Microbiology, Max Planck Institute for Terrestrial Microbiology, Marburg, Germany
2  LOEWE Center for Synthetic Microbiology (SYNMIKRO), Marburg, Germany
   *Corresponding author. Tel: +49 6421 2821475; E-mail: michal.skruzny@synmikro.mpi-marburg.mpg.de

and followed their rearrangements and binding dynamics by FRET and fluorescence recovery after photobleaching (FRAP), respectively. This system approach allowed us to draw a robust, high-resolution map of the endocytic coat. We found that endocytic coat proteins form several functional layers on both sites of the clathrin lattice. Moreover, key coat proteins transverse the lattice to connect the membrane and cytoplasmic areas of the coat. This design has extensive mechanistic and regulatory implications for the function of the endocytic coat during actin-driven endocytic vesicle formation.

# Results

## Comprehensive FRET-based protein proximity screen of the yeast endocytic coat

Although the size of the endocytic coat complex is below the resolution limit of current fluorescence microscopy, the *in vivo* organization of this supramolecular assembly can be efficiently assessed by FRET-based protein proximity mapping. To analyze proximities between individual fluorescently labeled coat-associated proteins, we used a robust and straightforward FRET technique, acceptor photobleaching. Here, a specific increase in fluorescence (dequenching) of the FRET donor attached to the first protein, occurring after photobleaching of the FRET acceptor on the second protein, reports the energy transfer and thus the close proximity between the proteins. We focused our proximity mapping on the late stage of endocytic coat formation with all known coat-associated proteins present (Weinberg & Drubin, 2012). For that, we stabilized endocytic coats on the flat plasma membrane by addition of Latrunculin A (LatA). LatA allows the full assembly of the endocytic coat, but blocks the subsequent actin polymerization-dependent membrane invagination steps of endocytosis (Kaksonen *et al*, 2003; Newpher *et al*, 2005; Kukulski *et al*, 2012).

We first constructed yeast strains expressing tandems of coat-associated proteins tagged with GFP and mCherry from their endogenous loci. Fusions of these fluorescent proteins were shown not to disturb coat protein expression and function (Kaksonen *et al*, 2003, 2005; Boeke *et al*, 2014; Dataset EV1), being a good FRET pair at the same time (Albertazzi *et al*, 2009; Aravamudhan *et al*, 2014). We C-terminally tagged 14 endocytic coat proteins conserved between yeast and humans (human homologs in parentheses): Syp1 (FCHo1/2); Ede1 (Eps15/R); Apl1 subunit of AP-2 complex (AP-2); Yap1801 and Yap1802 (CALM/AP180); Ent1 and Ent2 (epsins 1–3); Sla2 (Hip1R); Pan1 and Sla1 (intersectin-s); End3 (Eps15/R); clathrin subunits Chc1 and Clc1 (CHC, CLC); and Gts1 (small ArfGAP). We also tagged 4 conserved coat-associated regulators of actin polymerization at the endocytic site: Las17 (WASP/N-WASP), Vrp1 (WIP), Bzz1 (FCHSD1-2), and Lsb3 (SH3YL1). Additionally, we tagged selected coat-associated proteins on their N-termini with GFP in strains containing a partner protein with the C-terminal mCherry tag. The only combinations not assessed here were protein pairs tagged both N-terminally due to a non-functional N-terminal mCherry tag on our proteins. Altogether, we constructed 217 GFP-mCherry protein pair strains covering thus 91% of the 237 strains accessible by our system, which was limited by protein loss-of-function after tagging, the non-functional N-terminal mCherry tag,

and a suboptimal fluorescence signal of several low-abundant proteins (see Fig 1 and Dataset EV1 for details).

To estimate the maximal FRET values achievable in our system, we first analyzed FRET between the GFP-mCherry tandem C-terminally appended to Sla1 and Sla2 (Khmelinskii *et al*, 2012). We measured respective FRET efficiencies of $15.1 \pm 1.3\%$ and $18.8 \pm 1.4\%$ (mean FRET%, with 95% confidence intervals), which were indeed the highest values observed in our experiments.

We then analyzed FRET at the endocytic sites of LatA-treated cells expressing individual GFP/mCherry-tagged protein pairs. We detected 44 FRET-positive protein pairs with FRET efficiencies between 1.0 and 9.9% (Fig 1, Dataset EV1). The FRET microscopy data for 5 selected protein pairs are shown in Fig 2 and Movie EV1.

## Constructing the protein map of the endocytic coat

### Endocytic coat contains two separate protein proximity networks
We used the FRET proximities detected in our screen to construct the protein map of the endocytic coat in the direction perpendicular to the plasma membrane (Fig 3A; as discussed later, our screen does not provide sufficient information to build the coat map also in the plane parallel to the plasma membrane). Here, we used only two minimal constraints for the initial protein placement: First, it is thought that the N-terminal membrane-binding domains of Syp1, Apl1, Yap1801/2, Ent1/2, and Sla2 are adjacent to the plasma membrane. The N-terminus of Sla2, which can be functionally tagged with GFP (Picco *et al*, 2015), was shown to be placed ~1.5 nm from the membrane *in vitro* (Skruzny *et al*, 2015) and thus served as a topological marker for membrane proximity. Second, similarly to its human homolog Hip1R, Sla2 likely forms long dimers (Yang *et al*, 1999; Engqvist-Goldstein *et al*, 2001), with their C-termini being ~33 nm away from the plasma membrane (Picco *et al*, 2015).

When we connected the proximal termini of FRET-positive protein pairs, we observed three isolated pairs and two highly interlinked proximity networks (Fig 3A). The first proximity network (green in Fig 3A) connects the C-termini of Yap1801/2, Ent1/2, Sla1, Gts1, and End3 and the N-termini of Sla2, Gts1, Pan1, and End3. The second proximity network (red in Fig 3A) connects the N-termini of Sla1 and End3 and the C-termini of Sla2, Pan1, and End3. As these two proximity networks exclusively contained the opposite ends of extended modular proteins Sla2, Pan1, and Sla1, we hypothesized that their constituents occupy distinct areas of the endocytic coat.

### Endocytic coat proteins localize on both sites of the clathrin lattice
Next, we analyzed detected FRET proximities between endocytic coat proteins and the C-terminally tagged subunits of clathrin, Chc1 and Clc1 (orange in Fig 3A). Chc1 and Clc1 are positioned on opposite surfaces of the clathrin lattice, serving thus as topological markers for areas between the lattice and the plasma membrane (Chc1) and between the lattice and the cytoplasm (Clc1). The observed FRET between Chc1 and the C-termini of Apl1, Ent1/2, End3, Sla1, and Gts1 and the N-terminus of Pan1 strongly suggested that these constituents of the first proximity network are situated between the plasma membrane and the clathrin lattice.

The only proximity to Clc1 was found for the C-terminus of End3, with its FRET value being almost twice higher than for Chc1 (Fig 3B). Interestingly, two other proteins showed FRET with both End3 termini: The C-terminus of Sla1, a member of Chc1-proximal

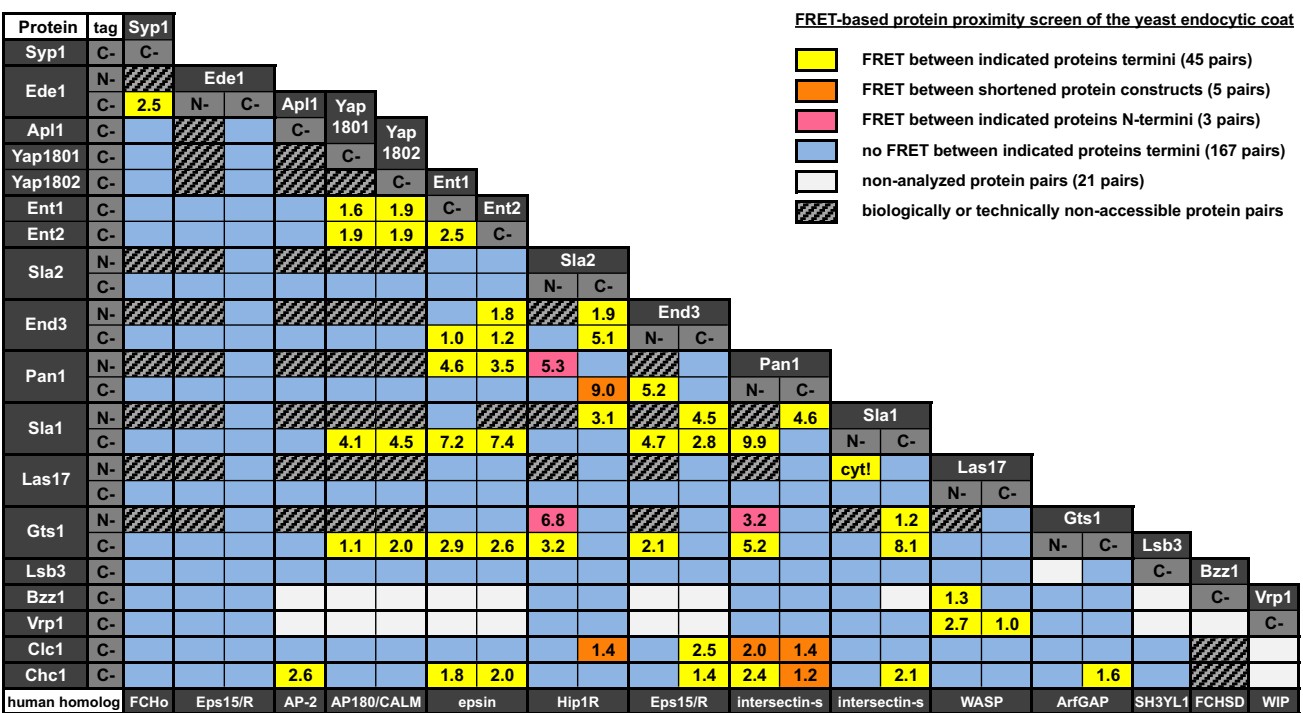

**Figure 1. Results of FRET-based protein proximity screen of the yeast endocytic coat.**

Mean FRET efficiencies (in %) are shown for FRET-positive protein pairs (yellow, orange, and magenta boxes). "cyt!" indicates FRET between Las17 and Sla1 proteins specifically detected in the cytoplasm. See legend on the top right for further details. For detailed results, see Dataset EV1.

Source data are available online for this figure.

network, showed higher FRET with the N-terminus than with the C-terminus of End3 (4.7 ± 0.8% versus 2.8 ± 0.6%). Inversely, the C-terminus of Sla2 showed lower FRET with the N-terminus than with the C-terminus of End3 (1.9 ± 0.6% and 5.1 ± 1.0%) (Fig 3B). Altogether, this was suggestive of the overall proximity of End3 to the clathrin lattice, with its N-terminus closer to the first proximity network, and its C-terminus closer to Clc1 and the second proximity network. Constituents of this network (the C-termini of Sla2 and Pan1, and the N-terminus of Sla1) could be thus too remote from the clathrin lattice to show FRET with Clc1.

To test this hypothesis, we measured FRET between clathrin subunits and C-terminally shortened Pan1 and Sla2. We deleted the C-terminal, 430-amino acid-long part of Pan1 [Pan1 (1–1050)], as well as the C-terminal ~7-nm-long THATCH domain of Sla2 (Brett *et al*, 2006). Both deletions were previously shown to retain essential functions, proper localization, and expression levels of respective wild-type proteins (Wesp *et al*, 1997; Skruzny *et al*, 2012; Bradford *et al*, 2015). Contrary to the full-length Pan1 and Sla2, we detected FRET of Pan1 (1–1050) with Clc1 and Chc1, and between Sla2dTHATCH and Clc1 (Fig 3B). This strongly indicates that the C-termini of Pan1 and Sla2, as well as the N-terminus of Sla1, are located at the cytoplasmic site of the clathrin lattice, being more than 10 nm apart from it.

### Key coat proteins transverse the clathrin lattice in the membrane–cytoplasm direction

We reasoned that protein shortening could be generally used to increase the precision of our proximity map. As the C-termini of

Sla2 and Pan1, both members of the cytoplasmic proximity network, showed no FRET, we measured FRET between the C-termini of Sla2 and Pan1 (1–1050) or a less shortened Pan1 (1–1303) construct. In both cases, we detected high FRET (9.0 ± 0.7% and 4.8 ± 0.5%). Lower FRET value in the later case suggested that the C-terminus of Pan1 is the most remote part of the cytoplasmic proximity network (Fig 3B).

Finally, we tested the proposed orientation of Pan1 by using a functional N-terminally shortened Pan1 (402–1480) construct (Bradford *et al*, 2015). Although FRET values between Chc1 and the N-termini of Pan1 (402–1480) or the full-length Pan1 were similar, only Pan1 (402–1480) showed FRET with Clc1, suggesting that this region is closer to the cytoplasmic site of the clathrin lattice (Fig 3B). These results strongly suggest the topology where both Pan1 and Sla2 have their N-termini between the plasma membrane and the clathrin lattice, their midparts span through the lattice, and their C-termini protrude deeply into the cytoplasm. Sla1 seems to have similar layout, but with the opposite orientation of its termini (Fig 3A and B).

### N-terminus of Gts1 is oriented towards the plasma membrane

The comparisons of FRET values of the N- and C-termini of Sla2, Pan1, Sla1, and End3 helped to orient them in the coat map. We applied this logic further to determine the orientation of ArfGAP protein Gts1 inside the membrane-proximal protein network. Although both C- and N-termini of Gts1 were well connected within this network, FRET between the C-terminus of Gts1 and the membrane-associated N-terminus of Sla2 indicated more membrane-

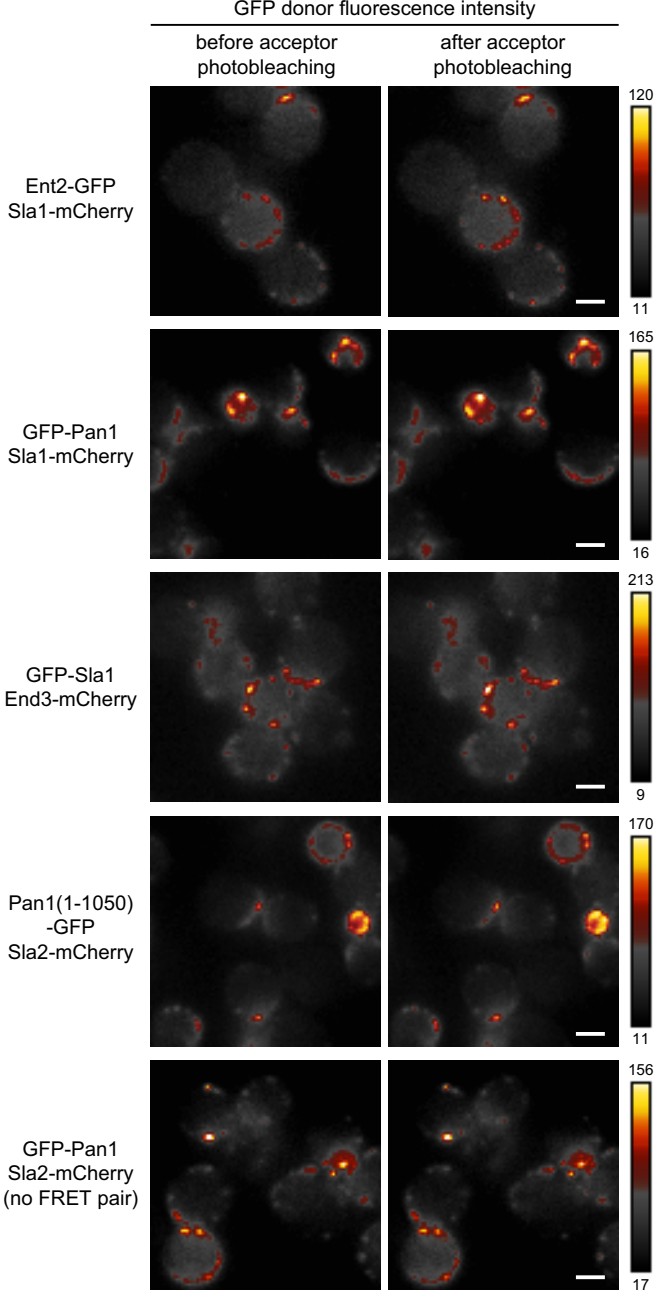

GFP donor fluorescence intensity

| before acceptor photobleaching | after acceptor photobleaching |

**Figure 2. Examples of FRET proximities between endocytic coat proteins.**

Fluorescence intensity of the FRET donor (GFP) is shown before and after photobleaching of the FRET acceptor (mCherry) in ImageJ Smart pseudocolor scheme (for intensity values, see bars on the right). FRET is observed as an increase in donor fluorescence after acceptor photobleaching. The indicated GFP- and mCherry-tagged endocytic proteins localize at the endocytic patches on the plasma membrane of live yeast cells. The enrichment of individual endocytic patches (diameter of 50–80 nm) on the plasma membrane, caused by the LatA-induced block of their subsequent invagination and/or their higher abundance at incipient buds, resulted in their apparent clustering on the diffraction-limited fluorescence images (pixel size, 178 nm). All raw acquisitions of GFP fluorescence intensities before and after photobleaching are shown in Movie EV1. Scale bars, 2 μm.

adjacent location of Gts1 (Fig 3C). Nevertheless, to locate Gts1 more precisely, we needed to test the proximity of its N-terminus to the N-termini of Pan1 and Sla2, which required a second functional N-terminal tag. This was achieved by introducing new FRET fluorophore pairs mTurquoise2/mNeonGreen and mNeonGreen/mScarlet-I, which can be both used for functional N-terminal tagging. We then measured FRET of both Gts1 termini tagged with mScarlet-I to the N-terminus of Pan1 or Sla2 tagged with mNeonGreen (Fig 3C). While the N-terminus of Pan1 showed similar FRET with both Gts1 termini (3.2 and 3.7%), the N-terminus of Sla2 showed much higher FRET with the N-terminus than with the C-terminus of Gts1 (6.7 ± 1.6% versus 2.8 ± 1.5%). In contrast, the C-terminus of Sla1 showed much higher FRET with the C-terminus of Gts1 than with its N-terminus (8.1 ± 0.9% versus 1.2 ± 0.6%). Together, these data strongly suggest that the N-terminus of Gts1 is placed closer to the plasma membrane than the C-terminus. Indirectly, these data also imply a more membrane-proximal location of the N-terminus of Pan1 compared with the C-terminus of Sla1, though probably still more distal than the N-termini of Gts1 and Sla2 (Fig 3A and C). Consistently, strong FRET was detected between the N-termini of Pan1 and Sla2 tagged with mNeonGreen and mScarlet-I.

### FRET proximity screen detects cytoplasmic assemblies of endocytic coat proteins

Several yeast endocytic coat-associated proteins were previously shown to interact already in the cytoplasm using fluorescence cross-correlation spectroscopy (FCCS; Boeke *et al*, 2014). We tested whether our FRET proximity screen could also identify these cytoplasmic subcomplexes. Indeed, we could detect cytoplasmic interactions of coat-associated proteins (Syp1-Ede1, Pan1-End3, and Las17-Sla1) previously found by FCCS (Boeke *et al*, 2014) by FRET and compared them with the FRET proximities of these proteins at endocytic sites (Table 1). Strikingly, while FRET between Syp1-Ede1 and Pan1-End3 pairs persisted at endocytic sites, FRET between Las17 and Sla1 was detected only in the cytoplasm. This was the case for C-terminally tagged Las17 and Sla1, for Las17 and Sla1 tagged with mTurquoise2 and mNeonGreen on the N- and C-terminus, respectively, or when both proteins were tagged on their N-termini (Table 1), close to the proposed Sla1-Las17 interaction surface (Feliciano & Di Pietro, 2012). These data indicate that a substantial rearrangement of the Sla1-Las17 interaction happens at the endocytic site already before the onset of actin polymerization.

### Rearrangements of coat proteins during membrane invagination

While in our screen we analyzed the endocytic coat on the flat plasma membrane, the reorganization of some parts of the coat was proposed to occur during membrane invagination both in yeast and mammals (Picco *et al*, 2015; Sochacki *et al*, 2017; Mund *et al*, 2018). To probe these nanometer-scale protein rearrangements, we measured FRET of selected proteins pairs at individual endocytic sites sorted according to their endocytic stage. For that, we additionally tagged protein Abp1 with mScarlet-I as a marker of actin-dependent invagination steps of endocytosis (Kukulski *et al*, 2012), and worked on fixed cells as recently established for super-resolution microscopy of yeast endocytosis (Mund *et al*, 2018).

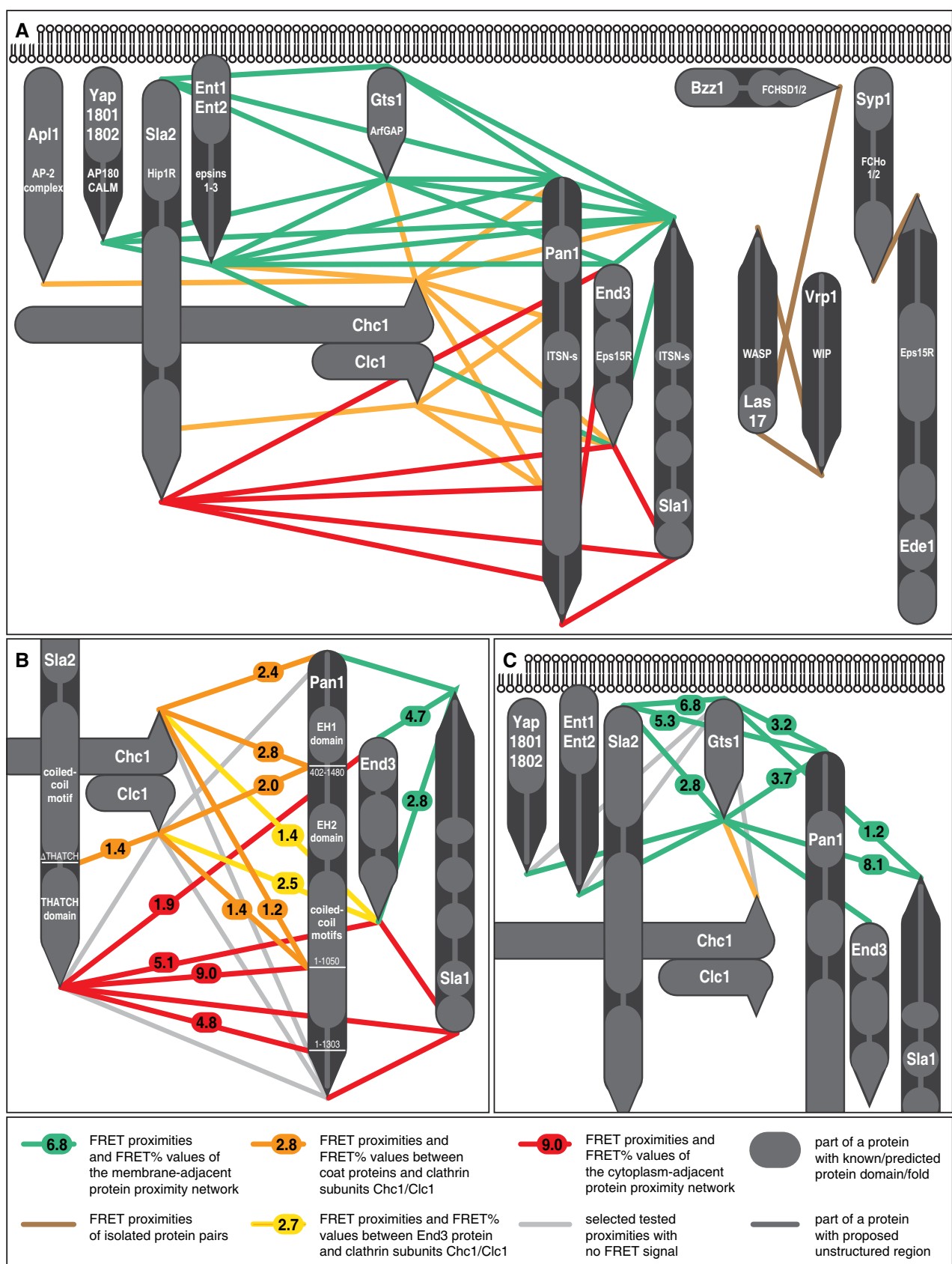

**Figure 3.**

**Figure 3.  Protein map of the yeast endocytic coat based on detected FRET proximities of coat-associated proteins.**

A   Protein proximity map of the endocytic coat-associated proteins.
B   Enlarged view of the map focused on selected proximities of clathrin subunits, End3, and Sla2 and Pan1 shortened and full-length proteins.
C   Enlarged view of the map focused on Gts1 proximity network.

Names of yeast proteins and their human homologs are shown in bigger and smaller letter sizes, respectively. Rounded and pointed ends of rods represent the N- and C-termini of indicated proteins, respectively. Single rods are used for Yap1801/2 and Ent1/2 protein duplicates, which have almost identical sets of proximity partners (see Dataset EV1). For color and shape coding, see legend at the bottom part of the figure. Numbers report mean FRET efficiencies (in %). Note that only single copy of each studied coat-associated protein [presented at the endocytic site in dozens of copies (Picco *et al*, 2015; Sun *et al*, 2019)] is depicted.

Source data are available online for this figure.

**Table 1.  Cytoplasmic assemblies of coat-associated proteins detected by FRET.**

| Protein pair[a] | | FRET%[b] | |
|---|---|---|---|
| | | Cytoplasm | Endocytic sites |
| Syp1• | Ede1• | 2.5 ± 0.6 (45) | 2.5 ± 0.8 (51) |
| •End3 | Pan1• | 4.5 ± 1.3 (29) | 5.2 ± 0.8 (30) |
| Las17• | Sla1• | 1.5 ± 2.2 (13) | 0.1 ± 0.7 (13) |
| •Las17 | Sla1• | 8.5 ± 1.3 (30) | 0.0 ± 2.3 (31) |
| •Las17 | •Sla1 | 9.8 ± 2.9 (24) | 0.7 ± 2.5 (36) |

[a]Syp1-GFP+Ede1-mCherry, GFP-End3+Pan1-mCherry, Las17-GFP+Sla1-mCherry, mTurquoise2-Las17+Sla1-mNeonGreen, and mTurquoise2-Las17 + mNeonGreen-Sla1 strains were used for the analysis. • denotes the position of the fluorophore tag on the protein.
[b]Mean FRET percentages with 95% confidence intervals and number of analyzed cells in parentheses are shown.

We focused on conceivable protein–protein rearrangements between the C-termini of Pan1 (1–1303)-Sla2, Ent1-Sla1, and Ent1-Sla2 tagged with mTurquoise2 and mNeonGreen. We hypothesized that during invagination the C-termini of Ent1 and Sla2 could colocalize in the cytoplasmic area of the coat due to their functional binding to the actin cytoskeleton (Skruzny *et al*, 2012), while the C-termini of Sla1 and Pan1 (1–1303) could separate to an outer rim of the coat as suggested recently (Picco *et al*, 2015; Sochacki *et al*, 2017; Mund *et al*, 2018) (Fig 4A). FRET between Pan1 (1–1303)-Sla2 was similar at the early endocytic sites that lack Abp1 and at invaginating sites that contain Abp1, suggesting that the C-termini of Pan1 (1–1303) and Sla2 remain proximal during membrane invagination (Fig EV1). In contrast, FRET between the C-termini of Ent1 and Sla1 was slightly, but significantly, decreased at the endocytic sites decorated by Abp1 (Fig 4B), indicative of partial separation of these termini during invagination. Most strikingly, although the C-termini of Ent1 and Sla2 showed no FRET at the early endocytic sites, high FRET between them was detected at invaginating endocytic sites (Fig 4C). This suggests that Ent1-Sla2 termini are initially separated but come into proximity during actin-dependent membrane invagination.

### Binding dynamics of coat-associated proteins at the endocytic site

Having the spatial organization of the endocytic coat resolved, we also wanted to analyze binding dynamics of its protein components. Specifically, we used fluorescence recovery after photobleaching (FRAP) to investigate whether coat-associated proteins are stably or dynamically bound at the endocytic site showing thus slow or fast exchange, respectively, of their in-site bleached molecules.

We followed FRAP of Ede1, Apl1, Yap1801, Gts1, Bzz1, Lsb3, Las17, and Vrp1 tagged with mNeonGreen similarly as it was done for Syp1, Yap1802, Ent1/2, Sla2, Pan1, Sla1, and End3 tagged with EGFP (Skruzny *et al*, 2012). Among studied proteins, Apl1, Yap1801, and Gts1 showed very low recovery, similar to previous findings for Yap1802, Ent1/2, Sla2, Pan1, Sla1, and End3 (Skruzny *et al*, 2012). Faster recovery was detected for Ede1, Las17, Lsb3, and also partly for Vrp1. Finally, the fastest recovery was observed for the F-BAR protein Bzz1 (Fig 5 and Movie EV2). In summary, FRAP experiments showed that while endocytic adaptors Apl1 and Yap1801, and Gts1 protein are stably bound at the endocytic coat, the endocytic priming proteins Ede1 and Syp1, and actin regulators Las17, Vrp1, Lsb3, and Bzz1 associate with the endocytic machinery in a dynamic manner.

## Discussion

To understand the mechanisms and regulation of endocytic vesicle formation, the knowledge of its underlying protein machinery is of utmost importance. Here, we determined the nanoscale organization of the endocytic coat using comprehensive FRET-based protein–protein proximity screen. We show that coat proteins are organized in distinct but interconnected functional layers situated on both sides of the clathrin lattice. Moreover, we detected rearrangements of the endocytic coat during membrane invagination and characterized binding dynamics of involved proteins.

We focused on the endocytic coat of budding yeast, the proteins of which are all, or almost all, known. In contrast to mammalian endocytic coats, which contain many lately evolved, auxiliary cargo-, tissue-, and splicing-specific factors, the yeast endocytic coat consists of only 22 proteins covering many important families of endocytic adaptors, scaffolds, and coat-associated actin regulators (Weinberg & Drubin, 2012; Goode *et al*, 2015; Lu *et al*, 2016). Notably, our 18 studied proteins are conserved in many species and potentially constitute the evolutionary core of the endocytic machinery (Dergai *et al*, 2016). The uncovered protein architecture of the endocytic coat is thus likely to be shared by many organisms.

The proposed organization of the yeast endocytic coat on the flat plasma membrane is depicted in Fig 6. Interacting endocytic pioneering proteins Ede1 and Syp1 (Reider *et al*, 2009) were found proximal to each other, but not to other proteins. This is probably a consequence of our focus on the late endocytic coat, where Ede1/Syp1 interactions can be already replaced by a stable protein network provided by similar domains of Pan1, End3, and Sla2

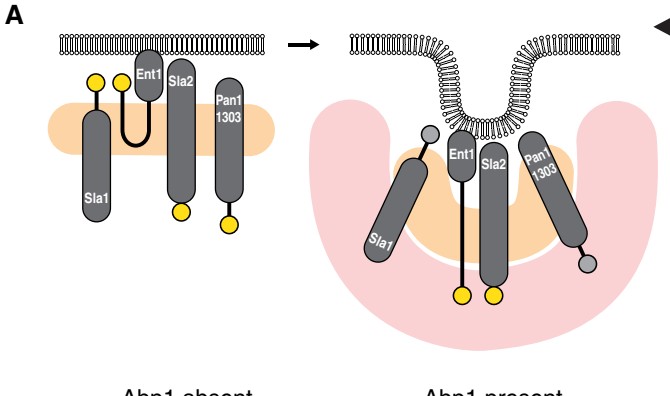

Abp1 absent        Abp1 present

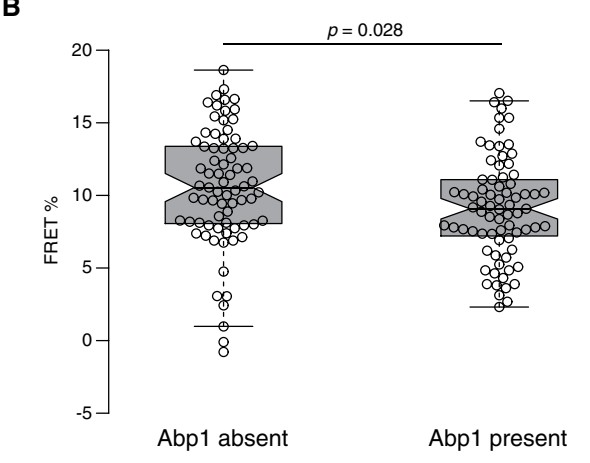

$p = 0.028$

Ent1-mTurquoise2 + Sla1-mNeonGreen

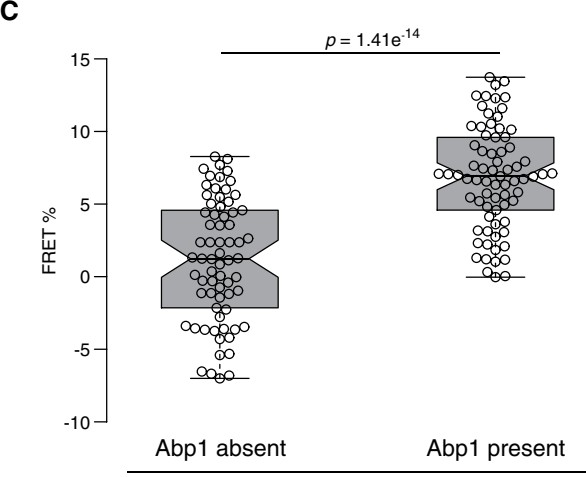

$p = 1.41e^{-14}$

Ent1-mTurquoise2 + Sla2-mNeonGreen

**Figure 4.**

**Figure 4.  Rearrangements of endocytic coat proteins during membrane invagination.**

A    Scheme of suggested topology of Sla1, Ent1, Sla2, and Pan1 (1–1303) proteins on the flat (left) and invaginated (right) plasma membranes. Natively unfolded parts of the proteins are shown as thick black lines. The C-termini of proteins showing FRET are depicted as yellow circles; the C-termini separated from their FRET partners during invagination are shown as gray circles. The position of the clathrin lattice and polymerizing actin cytoskeleton are shown as a pale orange and red zone, respectively.

B, C  Partial loss of FRET between Ent1-mTurquoise2 and Sla1-mNeonGreen (B) and presence of FRET between Ent1-mTurquoise2 and Sla2-mNeonGreen (C) during endocytic membrane invagination. FRET values (in %) of individual endocytic patches (*n* = 74, 75; 69, 73 for (B) and (C), respectively) sorted according to the absence/presence of Abp1 protein are shown as box plots. Center, top, and bottom lines of box plots show the medians, the 25th, and 75th percentiles of individual datasets, respectively. Whiskers extend 1.5 times the interquartile range from the 25th and 75th percentiles. Notches indicate 95% confidence intervals of the medians. Statistical differences were calculated by Welch's *t*-test. See also Fig EV1 and Appendix Fig S1.

Source data are available online for this figure.

mature coat as suggested by their recent super-resolution imaging (Mund *et al*, 2018) and their incomplete colocalization with other proteins in our screen (Dataset EV1).

As suggested by the established FRET proximities and low protein exchange in FRAP experiments (Figs 3 and 5; Skruzny *et al*, 2012), a stable structural core of the yeast endocytic coat is made of adaptors AP-2 (represented by Apl1 subunit), Yap1801/2, Ent1/2, and Sla2; and endocytic scaffolds Pan1, End3, Sla1, and ArfGAP protein Gts1. Proximity mapping with clathrin subunits and truncated Sla2 and Pan1 proteins as topology markers then sorted these factors into two distinct spatial layers located above and below the clathrin lattice. As expected, prototypical adaptor proteins AP-2, Yap1801/2, and Ent1/2 localize between the plasma membrane and clathrin, thus forming the endocytic adaptor layer. They share this region with Gts1, the N-termini of Sla2, Pan1, and End3, and the C-terminus of Sla1. FRET measurements with the shortened Sla2 and Pan1 proteins then established the second layer localized surprisingly on the cytoplasmic side of the clathrin lattice. It consists of the C-termini of Sla2, Pan1, and End3, and the N-terminus of Sla1. Sla2, Pan1, and Sla1 proteins therefore span through the clathrin lattice interconnecting both layers above and below it. As the cytoplasmic termini of Sla2, Pan1, and Sla1 are involved in either binding (Sla2, Pan1) or regulation (Pan1, Sla1) of the actin cytoskeleton at the endocytic site, we named this area the actin-regulatory layer. Based on FRET results of shortened Sla2 and Pan1 proteins, we propose that the actin-regulatory layer, and especially the C-terminus of Pan1, extend more than 10–20 nm from the clathrin lattice. Assuming that these distances are maintained during membrane invagination (Idrissi *et al*, 2012), Sla2, Pan1, and Sla1 are ideally placed not only for binding and regulation of the nascent actin cytoskeleton at the endocytic site, but also for the proposed communication of Pan1 and Sla1 with actin regulators Myo3/5, Las17, and Bbc1 localized at the outer, membrane-proximal ring around the coat (Barker *et al*, 2007; Idrissi *et al*, 2012; Picco *et al*, 2015; Mund *et al*, 2018).

We can also discern the part of the adaptor layer, consisting of the N-termini of Sla2 and Gts1, and likely all N-terminal membrane-binding domains of adaptors AP-2, Yap1801/2, and Ent1/2, as the

proteins. In agreement, our FRAP data show a similar dynamic association of Ede1 and Syp1 with the endocytic site, in contrast to a stable binding of Pan1, End3, and Sla2 (Fig 5; Skruzny *et al*, 2012). Ede1 and Syp1 are thus likely to dynamically localize outside of the

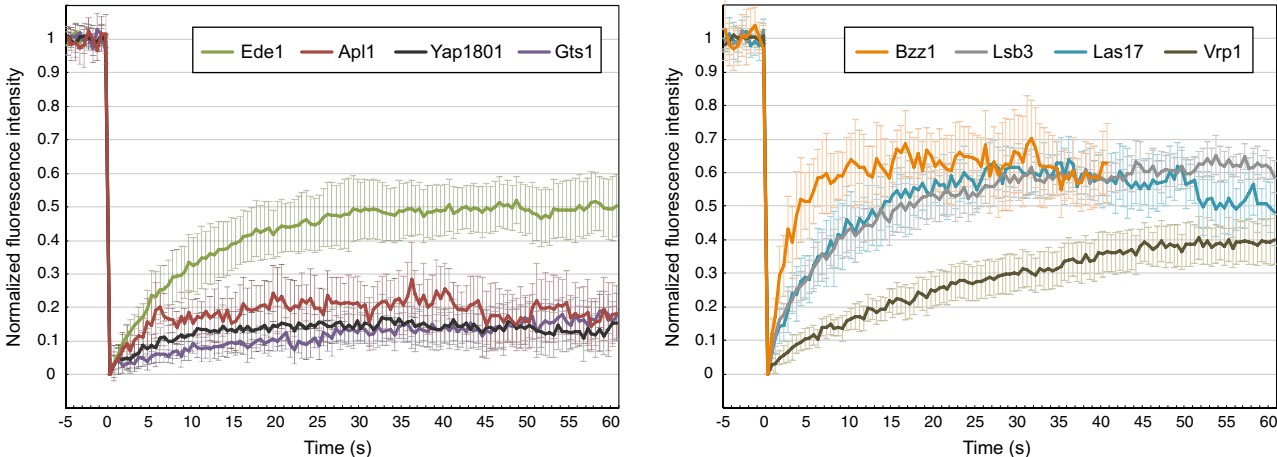

**Figure 5.   FRAP analysis of coat-associated proteins at the endocytic site.**

Ede1, Apl1, Yap1801, Gts1, Bzz1, Lsb3, Las17, and Vrp1 proteins tagged with mNeonGreen were photobleached in individual endocytic sites of LatA-treated cells, and fluorescence recovery was followed every 0.5 s for 60 s (40 s in case of the highly dynamic Bzz1 protein). The curves represent means ± 95% confidence intervals (*n* = 9–17). See also Movie EV2.

Source data are available online for this figure.

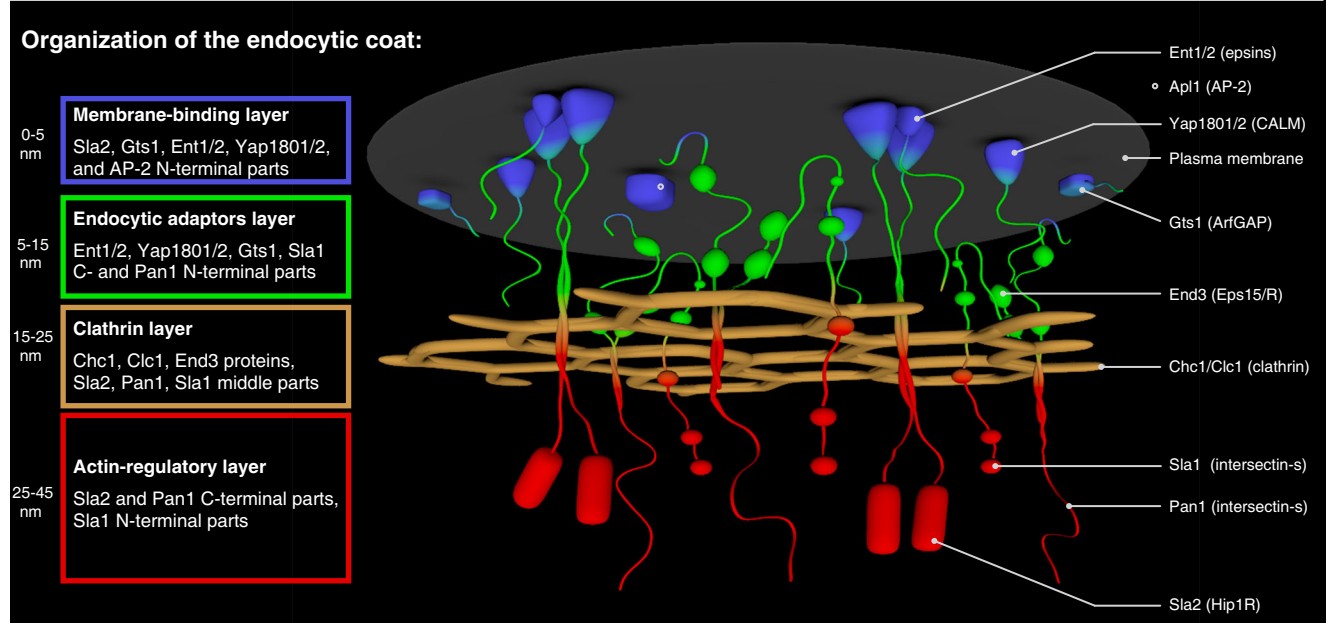

**Figure 6.   Model of the protein architecture of the endocytic coat.**

Individual coat layers together with their constituents and their approximate distances from the plasma membrane are listed on the left. Legend to individual depicted proteins (human homologs in parentheses) is shown on the right. Regulators of actin polymerization Las17, Vrp1, and Bzz1, and endocytic pioneering proteins Ede1 and Syp1, which were shown to localize in membrane-proximal regions outside of the depicted endocytic coat (Mund *et al*, 2018), are not shown for clarity. Note that only few copies of individual coat proteins (presented at the endocytic site in dozens of copies; Picco *et al*, 2015; Sun *et al*, 2019) are shown. See text for further details.

membrane-binding layer, which shows no FRET with clathrin. In agreement, the N-terminal parts of Gts1 contain ArfGAP domain and ALPS motif involved in membrane biochemistry of this protein (Smaczynska-de Rooij *et al*, 2008). The detected proximities of Gts1 thus provide a firm functional localization to this understudied endocytic regulator involved in the vesicle uncoating (Toret *et al*, 2008).

Finally, no proteins were apparently adjacent to the actin regulators Las17, Vrp1, and Bzz1, apart from the expected proximities between them (Naqvi *et al*, 1998; Soulard *et al*, 2002). This is consistent with recent evidence that these proteins localize in an outer ring outside the endocytic coat (Mund *et al*, 2018), where they bind, according to our FRAP studies, in a dynamic manner.

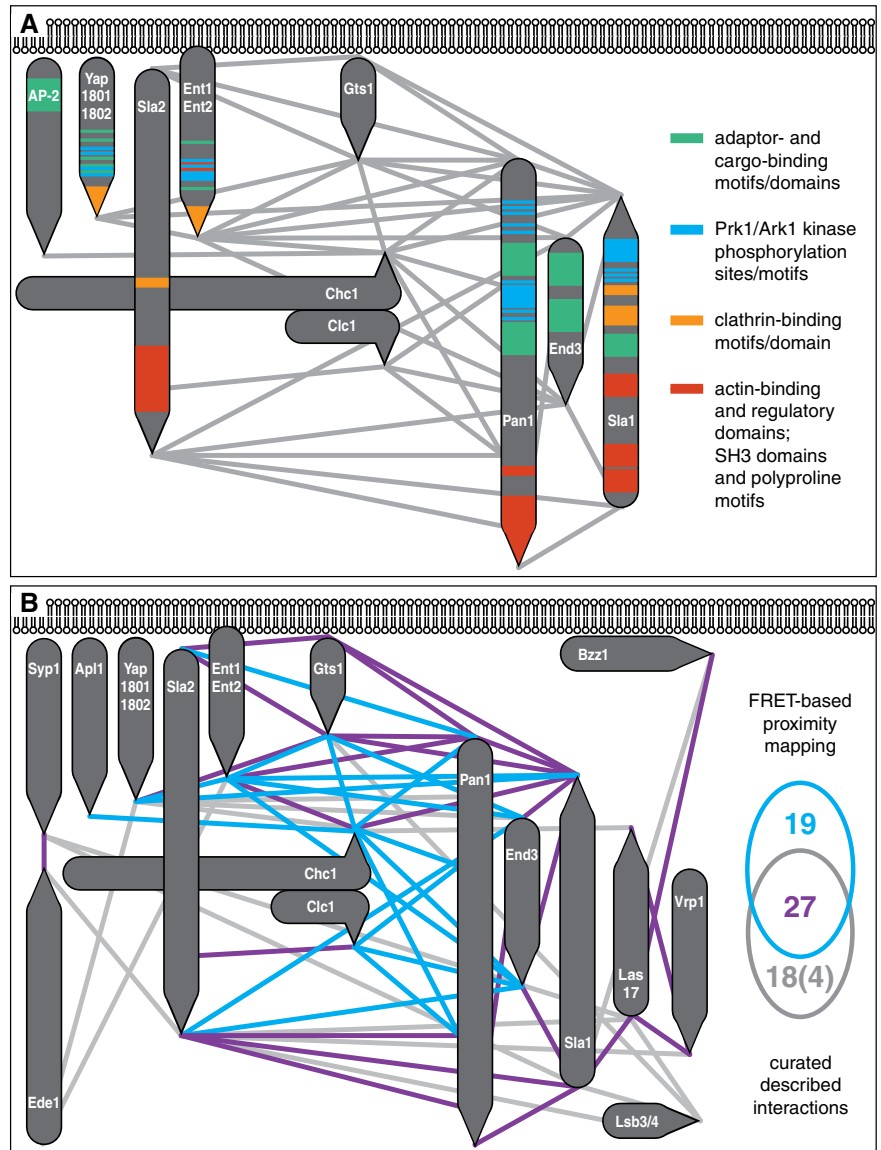

**Figure 7. Analysis of functional, binding, and regulatory moieties of the endocytic coat protein architecture.**

A   Topology of functional, binding, and regulatory motifs and domains in the proposed endocytic coat architecture. Known motifs and domains of indicated coat proteins were classified according to their main function and color-coded as shown on the right. Ark1/Prk1 phosphorylation sites were taken from (Ref. Huang *et al*, 2003).

B   Comparison of the FRET-based protein proximity mapping with previously described interactions of studied proteins. Physical interactions of indicated proteins (observed in more than one study) were taken from Saccharomyces Genome Database as per February 2019. FRET proximities mirroring previously detected protein interactions are shown as violet lines and numbers. Proximities not found as interactions and vice versa are shown as blue and gray lines/numbers, respectively. See the text for further details.

However, the lack of proximities between the coat proteins and the actin regulators leaves unanswered questions regarding the initiation of actin polymerization at the endocytic site. More specifically, while our detected cytoplasmic proximity between Sla1 and Las17 mirrors their previously described cytoplasmic complex (Feliciano & Di Pietro, 2012), supposedly blocking the Las17 activity in the cytoplasm, the absence of Sla1-Las17 proximity at the endocytic site complicates the current proposals on negative regulation of Las17 by Sla1 during endocytosis (Rodal *et al*,

2003; Sun *et al*, 2006). Further experiments are needed to elucidate this conundrum of timely activation of actin polymerization at the yeast endocytic site.

In addition, the absence of FRET between the coat proteins and proteins localized at the outer rings of the endocytic site (Syp1-Ede1, actin regulators) does not allow us to infer about a potential organization of the coat in the plane of the membrane. Based on the apparent homogeneity of the coat proteins at this plane shown by super-resolution studies (Mund *et al*, 2018), we assume that coat

proteins do not adopt very discrete organization along the flat membrane and the detected FRET proximities predominantly mirror their organization in the direction perpendicular to the plasma membrane (Fig 6).

On top of a comprehensive description of the endocytic coat architecture, our FRET experiments also identified two coat rearrangements occurring during membrane invagination. Firstly, we observed a decrease in FRET between the C-termini of Sla1 and Ent1, indicative of their partial separation. This is in line with recently suggested re-localization of Sla1, or its human homolog intersectin-s, to the outer rim of the coat during invagination (Picco et al, 2015; Sochacki et al, 2017). However, a mild reduction in FRET suggests that this Sla1-Ent1 segregation is either minor or occurs only within a sub-pool of Sla1-Ent1 molecules. Both notions are consistent with the recent super-resolution study showing that (i) only small changes (around 5 nm) of the average position of Sla1 partner Pan1 take place during endocytosis, and (ii) only a small fraction of endocytic sites shows the rim-organized Sla1 (Mund et al, 2018). Secondly, we observed the proximity between the C-termini of Ent1 and Sla2 during membrane invagination. The C-terminus of Ent1, which contains both clathrin- and actin-binding motifs (Wendland et al, 1999; Skruzny et al, 2012), thus seems to relocate from the adaptor layer of the coat to its actin-regulatory layer after the onset of the actin-driven membrane invagination. This supports the proposed role of the C-terminus of Ent1 in harnessing the force of actin polymerization for membrane invagination (Skruzny et al, 2012; Messa et al, 2014).

We also analyzed the acquired coat architecture with respect to known functional, binding, and regulatory motifs of the involved proteins (Fig 7A). As formation of the endocytic coat requires interaction of membrane-bound adaptors with clathrin, cargo, and each other, the exclusive presence of all known clathrin-, cargo-, and adaptor-binding motifs and domains in the adaptor and clathrin layers is perhaps not surprising. More intriguingly, all predicted phosphorylation sites of coat-associated kinase Ark1/Prk1 (AAK1/GAK1 in human) are also located in the adaptor layer, suggesting that this layer is the primary place of activity for these uncoating kinases. Notably, not only actin-binding and actin-nucleating domains of Sla2 and Pan1, respectively, but also Pan1/Sla1 polyproline motifs (PP) and Sla1 SH3 domains colocalize in the actin-regulatory layer of the coat. This is consistent with the proposed formation of a multivalent SH3-PP network made of Pan1, Sla1, and actin regulators Las17, Vrp1, Myo3/5, and Bbc1, necessary for an efficient regulation of actin polymerization at the endocytic site (Sun et al, 2017; Mund et al, 2018). Moreover, the cytoplasmic location of SH3-PP network is favorable for its potential role in the phase separation, recently suggested to aid in endocytic membrane invagination (preprint: Bergeron-Sandoval et al, 2018).

Finally, we compared the FRET-based endocytic coat map with known interactome of its protein constituents (Fig 7B). We found that our screen recapitulated and improved topology of 60% of previously described interactions between endocytic coat-associated proteins. As shown in Fig 7B, our analysis mostly missed interactions of Ede1, Syp1, Las17, and Lsb3 proteins, most probably because these factors are no longer in a close contact with the mature endocytic coat, which we focused on (see above). When these interactions were not included in the comparison, only 4 previously identified interactions were not recovered by our screen

(92% recovery; see Venn diagram in Fig 7B). Most importantly, the FRET-based proximity mapping generated a large and robust dataset of previously not reported 19 protein pair proximities, which may represent yet-to-be characterized functional protein–protein interactions or subcomplexes of the endocytic coat.

Altogether, our study provides a highly resolved spatial, regulatory, and functional map of the conserved endocytic coat. We believe that this comprehensive description will foster mechanistic understanding of the endocytic process and serve as a framework for characterization of other cellular multiprotein machines involved in membrane-remodeling or actin-based processes. In addition, we show FRET microscopy as a powerful tool to systematically analyze multiprotein cellular machines in vivo.

# Materials and Methods

### Yeast strains and media

Standard yeast media and protocols were used to manipulate yeast strains listed in Appendix Table S1. The N- and C-terminal tagging/truncation of yeast proteins was made by homologous recombination of respective genes with PCR cassettes previously described (Janke et al, 2004; Khmelinskii et al, 2011) or constructed in the laboratory. These cassettes also provide peptide linkers GS (GGGGS)$_3$ATENNS and RTLQVD inserted between a protein and its N- and C-terminal fluorescent tag, respectively. The additional DGGGGS linker (followed by RTLQVD) was provided by primers for C-terminal tagging of Ent1/2 and Yap1801/2 to avoid disruption of the C-terminal clathrin-binding motif of these proteins (Wendland et al, 1999). Binding of resulting Ent1/2-GFP fusions to clathrin was further confirmed by yeast two-hybrid assay (Appendix Fig S2; Collette et al, 2009). For microscopy, strains were grown to a log phase in a low fluorescence SD-Trp medium (prepared from LoFlo YNB; Formedium, UK). Cells were attached to Concanavalin A-coated (0.1 μg/ml, Sigma-Aldrich) 8-well glass slides (ibidi, Germany) and observed at 20°C. Where indicated, cells were incubated with 200 μM Latrunculin A (Enzo Life Sciences) for 10 min followed by 10–50 min of imaging. Cell fixation with formaldehyde (Thermo) was performed on cells attached to 8-well slides as described previously (Mund et al, 2018).

### FRET microscopy

Acceptor photobleaching was performed using a wide-field Eclipse Ti-E fluorescence microscope (Nikon, Japan) equipped with X-Cite Exacte LED light source, Perfect Focus System (PFS), and NIS-Elements AR software (4.40; Nikon). Images were acquired with Nikon 60× Plan Apo NA 1.45 oil immersion objective with 1.5× tube lens and iXon 897-X3 EMCCD camera (Andor) with EM gain set up to 250. Following filter sets were used to acquire mTurquoise2 (Ex 436/20, dc 455, Em 480/40), GFP (Ex 470/40, dc 495, Em 525/50), mNeonGreen (Ex 504/12, dc 520, Em 542/27), and mCherry/mScarlet-I (Ex 585/29, dc 605, Em 647/57) fluorescence (Chroma, Semrock). Two acceptor-channel and three to five donor-channel of 0.5- to 1-s acquisitions were taken before photobleaching of the acceptor by 3- to 5-s pulse of 150 mW 593 nm (mCherry/mScarlet-I) or 515 nm (mNeonGreen) solid-state laser (CNI, Changchun, China)

followed by three to five donor-channel and two acceptor-channel images. In general, a more abundant protein of a protein pair was used as the FRET acceptor. In addition, selected protein pairs were also analyzed with the opposite orientation of their fluorophore tags, giving qualitatively same FRET results as the original pairs (Dataset EV1). Also, FRET data of Ent1/Ent2 and Yap1801/Yap1802 proteins strongly suggest that our screen is robust against differences in donor and acceptor concentrations. Ent1/2 and Yap1801/2 are paralogs resulting from the whole-genome duplication of *Saccharomycotina* group. These proteins of similar composition, function, and presumably topology differ in their copy number at the endocytic site (Ent1, middle to high; Ent2, Yap1802 low to middle; Yap1801, very low). Though the number of molecules available for FRET differs between paralogs, both Ent1/2 and Yap1801/2 show identical sets of FRET proximities (see Dataset EV1).

Images were analyzed with ImageJ software (Schneider *et al*, 2012). Images were first subtracted of general background. Endocytic patches of each photobleached cell were then manually selected to accurately separate the fluorescence signal of studied proteins at endocytic patches from their fluorescence in the cytoplasm, where protein pairs were usually not in FRET proximity (see Table 1 for exceptions). The polygon selection tool was used to select all endocytic patches of cells in FRET proximity screen (Figs 1–3). The oval selection tool was used to select (i) individual endocytic patches for studies of coat rearrangements (Fig 4) and (ii) inner cytoplasmic (nucleus and vacuole free) areas for studies of cytoplasmic FRET (Table 1). FRET efficiency (calculated as percentage increase in donor fluorescence after acceptor photobleaching) was calculated using FRETCalc plugin (Stepensky, 2007) with the intensity threshold set up to the level of cytoplasmic fluorescence of the analyzed cell. Finally, FRET efficiency values were corrected by subtraction with FRET values of respective donor-only strains acquired in parallel (showing slightly negative FRET due to a mild structural photobleaching of the donor during acquisition; see Dataset EV1 for values).

**FRAP microscopy**

FRAP microscopy was performed using VisiFRAP System (Visitron Systems, Germany) build on Nikon Eclipse Ti-E microscope equipped with PFS, CoolLED pE-4000 light source, and operated by VisiView software (3.3.0.6). Images were acquired with Nikon 100× Apo TIRF NA 1.49 oil immersion objective and iXon Ultra-888 EMCCD camera (Andor). Chroma TIRF ET 514-nm filter set was used to acquire mNeonGreen fluorescence. Photobleaching was achieved by 20- to 100-ms pulse of 100 mW 515-nm laser (at 1–5% power), and fluorescence recovery was followed for 1–2 min with 500-ms frame rate. FRAP was analyzed as previously described (Skruzny *et al*, 2012).

**Statistics and reproducibility**

The sample sizes were selected based on previous studies with similar methodologies (Kaksonen *et al*, 2005; Picco *et al*, 2015; Sun *et al*, 2019). We performed 2–3 rounds of the FRET proximity screen with two independent yeast clones. Specifically, 122 strains covering all possible proximities between endocytic coat proteins, and all potential proximities between coat proteins and Ede1, Syp1, and actin

regulators Las17, Lsb3, Bzz1, and Vrp1 suggested from literature (gray lines in Fig 5) were screened three times. Strains of Ede1 and Syp1 proteins with low colocalization with a protein partner (35 strains) and protein pairs of actin regulators showing "no FRET" signal in the two initial rounds (60 strains) were screened twice. All other experiments were performed in three independent replicates. As data from different replicates were indistinguishable, they were pooled for statistical analyses. In the FRET screen of coat proteins (Figs 1–3), FRET values of 10–40 cells (approx. 80–200 endocytic patches) were used to calculate mean FRET efficiency and 95% confidence intervals (95% CI). Protein pairs were assigned as "FRET-positive" when their mean FRET and both 95% confidence intervals were positive values (see Dataset EV1). At least 30 cells were used to similarly calculate FRET values in the cytoplasm (Table 1). In the studies of coat rearrangements, FRET values of 69–74 individual Abp1-negative and 68–75 Abp1-positive patches were determined from at least 25 cells. Individual FRET measurements are then shown in Figs 4 and EV1 as box plots, including 95% CI notches. Experiment in Fig 4B was repeated twice with similar results (see Appendix Fig S1 for the second dataset). To calculate mean FRAP curves with 95% CI (Fig 5), single patches of 9–17 cells were analyzed. Two-tailed Welch's *t*-test for unpaired datasets of uneven variances was used to compare dataset pairs. All processed FRET and FRAP datasets are included in Source Data files to respective figures.

## Data availability

The datasets produced in this study are available in the following database: Image dataset: BioStudies S-BSST386.

**Expanded View** for this article is available online.

### Acknowledgements
We thank A. Zori Comba for 3D model in Fig 6; M. Abella Guerra and S. M. Murray for critical reading of the manuscript; and Prof. S. Lemmon (University of Miami) for *ENT1/2* and *CHC1* two-hybrid plasmids and strain. This work was funded by Deutsche Forschungsgemeinschaft (DFG) Research Grant SK 305/1-1.

### Author contributions
Conceptualization, project administration, MS; methodology and resources, MS, GM; investigation and formal analysis, MS, EP, SG; writing—original draft, MS; writing—review and editing, MS and VS; supervision and funding acquisition, MS and VS.

### Conflict of interest
The authors declare that they have no conflict of interest.

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
