## [Review Process File · Molecular Systems Biology]

The protein architecture of the endocytic coat analyzed by FRET microscopy

Michal Skruzny, Emma Pohl, Sandina Gnoth, Gabriele Malengo and Victor Sourjik

Review timeline:	Submission date:	17 th May 2019
	Editorial Decision:	26 th June 2019
	Revision received:	22 nd August 2019
	Editorial Decision:	11 th October 2019
	Revision received:	14 th February 2020
	Editorial Decision:	19 th March 2020
	Revision received:	2 nd April
	Accepted:	3 rd April

Editor: Jingyi Hou

Transaction Report:

1st Editorial Decision

26th June 2019

Thank you for submitting your work to Molecular Systems Biology. We have now heard back from the three reviewers who agreed to evaluate your manuscript. You will see from the comments below that the reviewers find the manuscript to be of interest. They raise however a series of concerns, which should be convincingly addressed in a revision.

Without reiterating all the points raised in the reviews below, some of the more substantial issues are the following:

- As indicated by reviewer #1, statistical report and experimental replicates need to be provided to better support the main conclusions. The relevant procedures should be clearly described in the main text.
- The concerns regarding the linker sequences and stoichiometry should be addressed. Reviewer #2 also provides constructive suggestions in this regard.

All other issues raised by the reviewers need to be satisfactorily addressed. As you may already know, our editorial policy allows in principle only a single round of major revision so it is essential to provide responses to the reviewers' comments that are as complete as possible. Please feel free to contact me in case you would like to discuss in further detail any of the issues raised by the reviewers.

REFeree REPORTS

Reviewer #1:

The manuscript by Skruzny et al. is devoted to elucidate the spatial organization of proteins in the assembly of clathrin coated pits CCP in yeast. The authors used a FRET microscopy-based "screen" to define 217 distances between coat-associated proteins in flat CCP and at different stages of invagination. The combinatorial analysis of protein couples revealed the spatial organization of proteins relative to each other and within the clathrin lattice. FRAP experiments demonstrated that some CCP-associated proteins are tightly bound to the lattice (Gts1, Yap1801, etc), whereas others are dynamic (Bzz1, Las17, etc).

The manuscript provides new interesting data on CCP organization that can of value to the cell biology community. The text is mostly well written and easy to read, although some parts are confusing (see specific comments below). The main general concern is the accuracy of the data to draw the conclusions, particularly the unclear statistics behind the data. I could not find from the description in the manuscript how many independent repeats of experiment were done. From the Methods one can find that the data were obtained on 2 clones with more than 6 cells per clone. Is it one independent experiment (cells seeding, treating and imaging)? The measured FRET efficiency is in the range of 2-9% with a claimed accuracy ~1%. This quantification of accuracy is a key point of this study. Unfortunately, one cannot be convinced by the reported values. How many patches per pair were analyzed per independent experiment? From the data presented (Fig.4B, C and Fig.EV2) one can conclude that FRET estimations of individual spots were pooled together from all measurements and the mean+-SEM was calculated. This approach is well known to be prone to underestimation of experimental error (see Vaux et al, EMBO reports, 2012). The data were corrected for cross-bleaching of donor by acceptor-bleaching illumination. However, the spread of cross-bleaching is not reported, and how much this influences accuracy of FRET is unclear.

Specific comments:

The authors claim that they "analyzed 227 pair-wise proximity of all conserved 18 coat-associated proteins" (Abstract and Introduction). This sentence is confusing. 18 proteins can have only $18 * 17 / 2 = 153$ pairs. Later in the Results section they state that they use 2 tags per protein on C and N terminus and 217 pairs cover 91% of possible combinations. But the two side tagging results in 612 pairs, then 217 pairs cover about 35% of possible combinations. It is clear that some couples did not work, but it has to be better explained.

The sentence on page 11 "By sorting the endocytic sites according the presence/absence of Abp1, we equipped the robust, but end-point acceptor photobleaching technique with the time resolution required for the analysis of protein rearrangements during endocytosis" is incomprehensible.

Fig.2 does not have scale bars on images.

For the FRET microscopy there is no information on pixel size. The number of pixels per endocytic spot is not reported.

The FRET intensity were measured in "manually selected by polygon or oval selection tool" area. To what extent does FRET accuracy depend on such manual selections? How much it depends on accuracy of background estimation? How much result depends on manual selections? All these issues must be clarified and reported in a much clearer fashion.

Reviewer #2:

In this paper, Skruzny and colleagues set out to clarify the molecular architecture of the coat complex that assembles during clathrin-mediated endocytosis. While prior studies have defined the core set of proteins that constitute the endocytic coat, determining the architecture of the assembled coat, as well as the relative position of individual proteins, has been a challenge, particularly in the context of living cells. This is in part because of the small size of individual endocytic sites (at or below the diffraction limit), the short lifetime of these sites (1-2 min), and the dynamic structural

arrangements that occur during this time.

The authors use an elegant FRET-based approach in live yeast cells, where it is possible to "trap" endocytic sites in which the core coat has assembled using Latrunculin A to prevent actin polymerization and completion of the endocytic event. They performed pairwise analysis of FRET efficiency between the majority of proteins that form the core coat in order to infer proximity of different proteins to each other (as well as proximity of both N- and C-termini for some proteins). Based on their results, they were able to assign proteins (or ends of proteins) to a membrane-proximal region, a region containing the clathrin lattice, and a membrane-distal region facing toward the cytoplasm. Of note, some proteins in the coat appear to adopt an extended configuration that passes through the clathrin lattice, and can connect the membrane-proximal and distal regions.

This study is an important body of work that provides a conceptual advance of our understanding of clathrin coat assembly and architecture, and as such should be of great interest. As outlined below, there are several areas of concern that would benefit from clarification and/or additional experimentation prior to publication. Once addressed, they will considerably strengthen the study, which should be very well received.

Major concerns:

1. The authors need to provide the precise amino acid sequence used for the linker between the coding sequence and the fluorescent tags. According to their methods, they used the PCR-based integration system and plasmids described by Janke et al. (2004, *Yeast* 21:947-62) and/or Khmelinskii et al. (2011, *PLoS ONE* 6:e23794), which itself is based upon the Janke et al. system.

The linker sequence is particularly important for Ent1, Ent2, Yap1801 and Yap1802. The authors correctly state that these four adaptors have a C-terminal clathrin-binding motif, LIDO* (O is a bulky, hydrophobic amino acid and * is the C-terminus), which closely resembles the consensus sequence for a type I clathrin box (LOXO[D/E]). Importantly, Wendland et al. (1999, *EMBO J* 18:4383-93) demonstrate that the carboxyl group at the extreme C-terminus of the epsins (and presumably the yeast AP180 homologs) substitutes for the acidic residue normally present at the final position of the consensus sequence (i.e., LIDL* of Ent1 binds to clathrin, but LIDLAAAA* does not). The aspartate residue in the center of the motif is not sufficient for binding to clathrin. Based on Janke et al., the first codon in the linker should be 5'-CGT-3', which corresponds to arginine. If this is indeed the first amino acid in the linker used for the C-terminally tagged adaptors here, it is predicted to disrupt the clathrin-binding motif.

There are other domains/motifs in the adaptors that act as localization determinants, including the ENTH/ANTH domain (membrane binding), NPF motifs that bind EH domain-containing proteins, and ubiquitin-interacting motifs that bind cargos, so the clathrin-binding motif likely is not required for recruitment of the adaptor proteins to endocytic sites and wouldn't necessarily affect lifetime, dynamics, etc. Skruzny et al. (2012, *PNAS* 109:E2533-42) did in fact show very nicely that Ent1-GFP localizes correctly, but in that paper there is again no information on the nature of the linker sequence. Notably, the C-terminal portion of adaptors is largely unstructured (i.e., Busch et al., *Nat Commun* 6:7875), and an intact/functional clathrin-binding motif likely plays an important role in correctly positioning this end of these proteins within the endocytic coat. If the first amino acid in the linker sequence is anything other than aspartate or glutamate, this raises the concern that all FRET-based proximity data involving any of the four adaptors do not reflect the true behavior of the endogenous proteins.

The above concerns don't necessarily apply if the authors used a linker in which the first amino acid is an aspartate or glutamate, in which case they should clarify this in the text, with appropriate explanation. If, however, the first amino acid is an arginine (or anything other than Asp/Glu), the authors should demonstrate that the tagged C-terminus of each adaptor can directly bind to clathrin *in vitro*, using highly purified proteins and not cell extracts that will also contain other endocytic proteins. Alternatively, they could make strains containing an acidic residue where the stop codon would normally be, though they should still perform the above binding experiments to demonstrate that the bulky fluorescent tag does not interfere with clathrin binding.

2. When using FRET to assess proximity of protein pairs (or ends of proteins), one might presume

that the stoichiometry of proteins would have an influence on the calculated FRET efficiency: if there are more FRET acceptor molecules, the efficiency might increase, and if there are more FRET donor molecules, it might decrease. Within the endocytic coat, the different proteins are not in a 1:1 stoichiometry; thus, FRET efficiency values might over- or under-represent whether pairs of proteins are in proximity. One way to determine whether this is the case is to test at least some strains where the donor and acceptor tags are on reciprocal proteins, particularly for pairs of proteins that are known to have different abundance at endocytic sites. If stoichiometry does not factor into the calculated efficiency, flipping the tags should give the same values. Have the authors tested this?

3. In Figure 4, the authors examine FRET efficiency between Ent1-mTurquoise2 and Sla1-mNeonGreen in cells that also express Abp1-mScarlet1. Of note, they also use mNeonGreen and mScarlet1 as a FRET pair in earlier experiments. While it seems unlikely that the presence of two FRET pairs simultaneously (both of which include mNeonGreen) would have a major impact on the calculated FRET values, have the authors tested whether this is actually the case by using a strain in which Abp1 is tagged with something that is not a FRET pair with either of the other two tagged proteins?

Minor concerns

4. In the discussion, the authors correctly indicate that there are 22 proteins in the conserved core endocytic coat, but in the abstract they state that they tested "all" 18 conserved proteins. At least some of the proteins not tested (Apm4/ μ -2) are also conserved

5. Figure 2 should include a scale bar.

6. Figure 4 was difficult to interpret. The x-axes for panels B and C indicate which pair of proteins were analyzed, rather than stating Abp1 absent (left column) or Abp1 present (right column). This information was indicated above panel B to link the graphs to the diagram above, but is counterintuitive to a reader.

7. The FRAP data (last part of the results and Fig. 5) don't really seem to go with, or add to, the remainder of the study using FRET. The authors might want to add more context to this part of the study.

8. Also related to the FRAP data, what are the $t(1/2)$ and mobile/immobile fractions for the proteins analyzed? Based on the recovery curves shown, the slow recovery rate reported for Apl1, Yap1801 and Gts1 might also be interpreted as a very low mobile fraction, where the small portion of the protein that is part of the mobile fraction actually recovers quickly. Slow recovery would imply a reduced rate that eventually recovers to the same final value, but these curves seem to plateau quickly, though to a greatly reduced value.

9. In Fig. 3, the use of red/yellow/green to indicate different proximity networks may be difficult to interpret for individuals who cannot distinguish between these colors.

10. What is the cutoff value for "no FRET" in this study? On pp.6-7, a range of 1.1-8.9% is implied, but in Table 1, there are some non-zero results that are not reported in Fig. 1. It would be helpful to explicitly define the cutoff value.

Reviewer #3:

This is an interesting study of the architecture of the clathrin-coated membrane formed in preparation for endocytosis in budding yeast. Technically the work seems to be well done and the authors are careful not to over-interpret their FRET efficiencies as distances. However, some aspects of the interpretation need to be modified, because the authors tend to think in one dimension, while their signals are coming from a 3D network. The interpretative figures also need extensive revision to be more anatomically correct, to take into account stoichiometries and to be more informative.

Page 6: Start new paragraphs with "To estimate.." and "We then analyzed..."

Figure 2: State whether the cells are alive or fixed. Provide a better anatomical description of the micrographs. For example, note that most of the spots are clusters of endocytic patches rather than single patches. Please explain why studying such closely packed patches is not a problem. Explain how GFP and mCherry are linked in the tandem construct.

Page 7: Figure 3 is a well-intentioned approach to illustrate the interaction network based on the FRET observations, but this figure is more confusing than edifying. The grey bars are inadequate representations of the protein molecules and not explained. For example, what do the lengths of bars represent? Obviously, it is not the sizes of the proteins, given the way Chc and Clc are depicted. How about taking better advantage of the known structure of clathrin and other proteins to illustrate the findings with scale drawings? For example, a scale drawing of the clathrin molecule (from the coated vesicle atomic model) would show that it extends across the entire figure a known distance from the lipid bilayer. Where other structures or domains are known, they should be depicted in proportion to their sizes and properly oriented relative to the lipid bilayer. For example, the BAR domain of Bzz1 should be horizontal on the bilayer. Consider including the common names of these proteins in the figure, since the gene names can confuse people in the field and are hopeless for outsiders. Consider taking advantage of the UCSF "Modeler" suit of programs to build this figure. Please include directly in the panels the color code for the lines.

Two other issues should be addressed:

(1) How to take into account both the X-Y and Z distances between the FPs? The text on page 9 concludes that "the N-terminus of Sla1 is located more than 10 nm from the clathrin lattice." But, is this 10 nm in Z or in X-Y or some combination of the two. The same problem occurs for each protein pair in the figures and throughout the text. One example of apparent separation in the X-Y plane is the absence of signal between the Bzz1 F-BAR domain and the green network. Again, on page 10, only Z is considered in concluding "a more membrane proximal location of the N-terminus of Pan1..." The same applies to the interpretation of the data in Figure 4.

(2) The stoichiometry of the components. Figure 3 has one copy of each protein, but Figure 6 has multiple copies. Is either depiction accurate?

Page 9: Deletion of the C-terminus of fission yeast Pan1 compromises actin patch assembly (Chen, Curr Biol, 2013), so I am skeptical that the Sc Pan1 truncation is fully functional.

Page 11: What is the meaning of "previously described cytoplasmic interactions of coat-associated proteins?" Is this FRET signal coming from the dispersed phase of the cytoplasm or from the cytoplasm adjacent to a site of endocytosis?

Figure 4: The relative scales of the membrane invagination, clathrin coat, actin network and the proteins are incorrect by more than an order of magnitude, so the depiction is misleading.

The discussion is simply a restatement of the data and interpretations already presented in the results section. This redundancy is not necessary. Perhaps all of the interpretation can be moved to the discussion.

1st Revision - authors' response

22nd August 2019

We thank all reviewers for their valuable and helpful comments and provide our point-by-point response below.

Reviewer #1:

The manuscript by Skruzny et al. is devoted to elucidate the spatial organization of proteins in the assembly of clathrin coated pits CCP in yeast. The authors used a

FRET microscopy-based "screen" to define 217 distances between coat-associated proteins in flat CCP and at different stages of invagination. The combinatorial analysis of protein couples revealed the spatial organization of proteins relative to each other and within the clathrin lattice. FRAP experiments demonstrated that some CCP-associated proteins are tightly bound to the lattice (Gts1, Yap1801, etc), whereas others are dynamic (Bzz1, Las17, etc).

The manuscript provides new interesting data on CCP organization that can of value to the cell biology community. The text is mostly well written and easy to read, although some parts are confusing (see specific comments below). The main general concern is the accuracy of the data to draw the conclusions, particularly the unclear statistics behind the data. I could not find from the description in the manuscript how many independent repeats of experiment were done. From the Methods one can find that the data were obtained on 2 clones with more than 6 cells per clone. Is it one independent experiment (cells seeding, treating and imaging)?

The measured FRET efficiency is in the range of 2-9% with a claimed accuracy ~1%. This quantification of accuracy is a key point of this study. Unfortunately, one cannot be convinced by the reported values. How many patches per pair were analyzed per independent experiment? From the data presented (Fig.4B, C and Fig.EV2) one can conclude that FRET estimations of individual spots were pooled together from all measurements and the mean+SEM was calculated. This approach is well known to be prone to underestimation of experimental error (see Vaux et al, EMBO reports, 2012). The data were corrected for cross-bleaching of donor by acceptor-bleaching illumination. However, the spread of cross-bleaching is not reported, and how much this influences accuracy of FRET is unclear.

As in vivo FRET signal usually constitutes a few percent of total detected fluorescence, we were maximally careful to do not over-interpret our data and support them with proper statistics. We now detail all statistics (e.g. number of independent experiments, analyzed patches, used statistical tests) in the separate section of Materials and Methods "Statistics and reproducibility".

Shortly here, in the FRET proximity screen two yeast clones were imaged in two independent sessions contributing by 40+ patches and giving similar results. FRAP and FRET studies of protein rearrangements were performed 3 times with 3+ patches/cells and 22+ patches (8+ cells) per session, respectively. The Fig. 4 and EV1 show FRET values of all patches with comprehensive statistics depicted as box plots (including 95% confidence intervals and p-values of Welch's t-test). In general, 95% confidence intervals (95% CI), not SEM, were used to indicate variance of individual datasets. For all figures, the original measured data are included as Source Data files.

The mean FRET values of donor-only strains (used to correct for a general photobleaching) and their 95% CI are now included in Dataset EV1.

Specific comments:

The authors claim that they "analyzed 227 pair-wise proximity of all conserved 18 coat-associated proteins" (Abstract and Introduction). This sentence is confusing. 18 proteins can have only $18 * 17 / 2 = 153$ pairs. Later in the Results section they state that they use 2 tags per protein on C and N terminus and 217 pairs cover 91% of possible combinations. But the two side tagging results in 612 pairs, then 217

pairs cover about 35% of possible combinations. It is clear that some couples did not work, but it has to be better explained.

We corrected respective sentences in the Introduction and Results:

(page 4, par.3) We analyzed 227 pairwise **protein-protein** proximities between the **N- and/or C-termini** of **all** 18 conserved coat-associated proteins at the yeast endocytic site...

(page 6, end of par.1) Altogether, we constructed 217 GFP-mCherry protein pair strains covering thus 91% of the 237 strains **accessible by our system, which was limited by protein loss-of-function after tagging, the non-functional N-terminal mCherry tag, and suboptimal fluorescent signal of several low-abundant proteins.** (see Fig 1 and Dataset EV1 for details).

The sentence on page 11 "By sorting the endocytic sites according the presence/absence of Abp1, we equipped the robust, but end-point acceptor photobleaching technique with the time resolution required for the analysis of protein rearrangements during endocytosis" is incomprehensible.

We thank the reviewer for this point. Simple deletion of the sentence made the text easier to read without losing any substantial information.

Fig.2 does not have scale bars on images.

We now provide scale bars in Fig. 2.

For the FRET microscopy there is no information on pixel size. The number of pixels per endocytic spot is not reported.

The information is now provided in the legend of Fig. 2.

The FRET intensity were measured in "manually selected by polygon or oval selection tool" area. To what extent does FRET accuracy depend on such manual selections? How much it depends on accuracy of background estimation? How much result depends on manual selections? All these issues must be clarified and reported in a much clearer fashion.

As fluorescent proteins at the endocytic sites appear as diffraction-limited spots and clusters at the cell cortex (Fig. 2), an accurate spot/cluster selection is important to specifically follow fluorescent changes in these areas, hence omitting the fluorescence of the surrounding cytoplasm. An inclusion of cytoplasmic signal will decrease the overall sensitivity of FRET detection, giving eventually false negative results, because a vast majority of tested proteins is also present in the cytoplasm, but not proximal to each other there. (Inversely, in case of Sla1-Las17 pair forming the complex specifically in the cytoplasm, inclusion of cytoplasmic regions will give a false positive result.)

We used manual selection tools to precisely include all patches of photobleached cells as this was not reliably possible by any automatic segmentation algorithm

tested. The polygon tool was used to select all endocytic patches and clusters of cells in the FRET proximity screen (Figs.1-3). The oval tool was used to select i) individual endocytic patches for studies of coat rearrangements (Fig. 4) and ii) cytoplasmic (nucleus and vacuole free) areas for studies of cytoplasmic FRET proximities (Table 1).

We agree with the reviewer that these procedures should be more clearly described, which we now made in Materials and Methods section “FRET Microscopy”.

Thresholding the signal of endocytic patches to the level of surrounding cytoplasmic background is the most correct way to calculate its fluorescent changes, nevertheless FRET results w/o thresholding were very similar and can be eventually provided (being also the outcome of used FRETcalc plugin).

Reviewer #2:

In this paper, Skruzny and colleagues set out to clarify the molecular architecture of the coat complex that assembles during clathrin-mediated endocytosis. While prior studies have defined the core set of proteins that constitute the endocytic coat, determining the architecture of the assembled coat, as well as the relative position of individual proteins, has been a challenge, particularly in the context of living cells. This is in part because of the small size of individual endocytic sites (at or below the diffraction limit), the short lifetime of these sites (1-2 min), and the dynamic structural arrangements that occur during this time.

The authors use an elegant FRET-based approach in live yeast cells, where it is possible to "trap" endocytic sites in which the core coat has assembled using Latrunculin A to prevent actin polymerization and completion of the endocytic event. They performed pairwise analysis of FRET efficiency between the majority of proteins that form the core coat in order to infer proximity of different proteins to each other (as well as proximity of both N- and C-termini for some proteins). Based on their results, they were able to assign proteins (or ends of proteins) to a membrane-proximal region, a region containing the clathrin lattice, and a membrane-distal region facing toward the cytoplasm. Of note, some proteins in the coat appear to adopt an extended configuration that passes through the clathrin lattice, and can connect the membrane-proximal and distal regions.

This study is an important body of work that provides a conceptual advance of our understanding of clathrin coat assembly and architecture, and as such should be of great interest. As outlined below, there are several areas of concern that would benefit from clarification and/or additional experimentation prior to publication. Once addressed, they will considerably strengthen the study, which should be very well received.

Major concerns:

1. The authors need to provide the precise amino acid sequence used for the linker between the coding sequence and the fluorescent tags. According to their methods, they used the PCR-based integration system and plasmids described by Janke et al.

(2004, *Yeast* 21:947-62) and/or Khmelinskii et al. (2011, *PLoS ONE* 6:e23794), which itself is based upon the Janke et al. system.

The linker sequence is particularly important for Ent1, Ent2, Yap1801 and Yap1802. The authors correctly state that these four adaptors have a C-terminal clathrin-binding motif, LIDO* (O is a bulky, hydrophobic amino acid and * is the C-terminus), which closely resembles the consensus sequence for a type I clathrin box (LOXO[D/E]). Importantly, Wendland et al. (1999, *EMBO J* 18:4383-93) demonstrate that the carboxyl group at the extreme C-terminus of the epsins (and presumably the yeast AP180 homologs) substitutes for the acidic residue normally present at the final position of the consensus sequence (i.e., LIDL* of Ent1 binds to clathrin, but LIDLAAAA* does not). The aspartate residue in the center of the motif is not sufficient for binding to clathrin. Based on Janke et al., the first codon in the linker should be 5'-CGT-3', which corresponds to arginine. If this is indeed the first amino acid in the linker used for the C-terminally tagged adaptors here, it is predicted to disrupt the clathrin-binding motif.

There are other domains/motifs in the adaptors that act as localization determinants, including the ENTH/ANTH domain (membrane binding), NPF motifs that bind EH domain-containing proteins, and ubiquitin-interacting motifs that bind cargos, so the clathrin-binding motif likely is not required for recruitment of the adaptor proteins to endocytic sites and wouldn't necessarily affect lifetime, dynamics, etc. Skruzny et al. (2012, *PNAS* 109:E2533-42) did in fact show very nicely that Ent1-GFP localizes correctly, but in that paper there is again no information on the nature of the linker sequence. Notably, the C-terminal portion of adaptors is largely unstructured (i.e., Busch et al., *Nat Commun* 6:7875), and an intact/functional clathrin-binding motif likely plays an important role in correctly positioning this end of these proteins within the endocytic coat. If the first amino acid in the linker sequence is anything other than aspartate or glutamate, this raises the concern that all FRET-based proximity data involving any of the four adaptors do not reflect the true behavior of the endogenous proteins.

The above concerns don't necessarily apply if the authors used a linker in which the first amino acid is an aspartate or glutamate, in which case they should clarify this in the text, with appropriate explanation. If, however, the first amino acid is an arginine (or anything other than Asp/Glu), the authors should demonstrate that the tagged C-terminus of each adaptor can directly bind to clathrin *in vitro*, using highly purified proteins and not cell extracts that will also contain other endocytic proteins. Alternatively, they could make strains containing an acidic residue where the stop codon would normally be, though they should still perform the above binding experiments to demonstrate that the bulky fluorescent tag does not interfere with clathrin binding.

We appreciate the reviewer for this important point. We had been aware of a disruption of the C-terminal clathrin-binding motif of Ent1/2 by tagging from early on (see e.g. SI Table 2 in Skruzny et al. 2012). For this study we indeed specifically provided additional GAT codon for aspartate (followed by codons for neutral GGGGS linker) on primers to be placed between the final codon of ENT1, ENT2, YAP1801, YAP1802 genes and the first codon of the cassette-derived linker

RTLQVD. We now describe this in Material and Methods section “Yeast strains and media”.

Our FRET analyses show that all four C-termini of Ent1/2 and Yap1801/2 fusions localize at the proximity of clathrin either directly (specific FRET between Ent1/2 and Chc1) or indirectly (localization of Yap1801/2 in the clathrin-adjacent “endocytic adaptors layer”), being thus presumably proficient of clathrin binding. We wish to exempt from performing suggested confirmatory in vitro binding studies as we are currently not established for protein purifications and know from past that the successful purification of these modular, natively unfolded proteins requires a highly experienced approach.

2. When using FRET to assess proximity of protein pairs (or ends of proteins), one might presume that the stoichiometry of proteins would have an influence on the calculated FRET efficiency: if there are more FRET acceptor molecules, the efficiency might increase, and if there are more FRET donor molecules, it might decrease. Within the endocytic coat, the different proteins are not in a 1:1 stoichiometry; thus, FRET efficiency values might over- or under-represent whether pairs of proteins are in proximity. One way to determine whether this is the case is to test at least some strains where the donor and acceptor tags are on reciprocal proteins, particularly for pairs of proteins that are known to have different abundance at endocytic sites. If stoichiometry does not factor into the calculated efficiency, flipping the tags should give the same values. Have the authors tested this?

The reviewer is fully right with her/his proposal and we indeed measured FRET of several protein pairs with flipped donor and acceptor tags to control for this phenomenon. We newly highlight these pairs in Dataset EV1 by asterisks. Pairs with flipped fluorophores always showed the same qualitative outcome (FRET/no FRET), in case of FRET with very similar values. Moreover, our qualitatively identical FRET results of protein duplicates Ent1/2 or Yap1801/2 (see, please, Fig. 1 and Dataset EV1), which differ substantially in their expression levels (but presumably not in their topology), provide an additional support for a robustness of our screen against concentration differences between donor and acceptor proteins (both described in Material and Methods, FRET Microscopy).

3. In Figure 4, the authors examine FRET efficiency between Ent1-mTurquoise2 and Sla1-mNeonGreen in cells that also express Abp1-mScarlet1. Of note, they also use mNeonGreen and mScarlet1 as a FRET pair in earlier experiments. While it seems unlikely that the presence of two FRET pairs simultaneously (both of which include mNeonGreen) would have a major impact on the calculated FRET values, have the authors tested whether this is actually the case by using a strain in which Abp1 is tagged with something that is not a FRET pair with either of the other two tagged proteins?

The reviewer is correct: Abp1-mScarlet could be eventually an acceptor for Sla1-mNeonGreen in our system and increase thus its acceptor propensities towards Ent1-mTurquoise2 (leading theoretically to a slight underestimation of the observed drop in FRET). Unfortunately, there is no suitable red-shifted fluorophore with “no FRET” spectral characteristics to mNeonGreen available in yeast to test this proposal.

Nevertheless, we also used Rvs167-mScarlet-I protein (potentially quite remote of studied acceptor proteins Sla1/Sla2) as a marker of non- and mildly invaginated vs. deeply invaginated endocytic sites. The FRET changes between Rvs167-absent and Rvs167-present sites were very similar to changes observed with Abp1 temporal marker (data not shown).

Minor concerns

4. In the discussion, the authors correctly indicate that there are 22 proteins in the conserved core endocytic coat, but in the abstract they state that they tested "all" 18 conserved proteins. At least some of the proteins not tested (Apm4/ μ -2) are also conserved

We took Apl1 subunit as a representative of AP-2 heterotetrameric complex, though all subunits are indeed conserved. We also did not work with Lsb4/Ysc84 genomic duplicate of Lsb3, which is not detectably expressed to our strain (commented in Dataset EV1). To do not confuse readers we removed the word "all" from respective places in the text.

5. Figure 2 should include a scale bar.

We now provide scale bars in Fig. 2.

6. Figure 4 was difficult to interpret. The x-axes for panels B and C indicate which pair of proteins were analyzed, rather than stating Abp1 absent (left column) or Abp1 present (right column). This information was indicated above panel B to link the graphs to the diagram above, but is counterintuitive to a reader.

We remade Fig. 4 as suggested.

7. The FRAP data (last part of the results and Fig. 5) don't really seem to go with, or add to, the remainder of the study using FRET. The authors might want to add more context to this part of the study.

We now provide a new introductory sentence to better tight this part with the other text.

8. Also related to the FRAP data, what are the $t(1/2)$ and mobile/immobile fractions for the proteins analyzed? Based on the recovery curves shown, the slow recovery rate reported for Apl1, Yap1801 and Gts1 might also be interpreted as a very low mobile fraction, where the small portion of the protein that is part of the mobile fraction actually recovers quickly. Slow recovery would imply a reduced rate that eventually recovers to the same final value, but these curves seem to plateau quickly, though to a greatly reduced value.

In our experimental setup we cannot easily discern between the suggested mix of very small, highly mobile and large, immobile fraction against (almost) completely immobile single pool. An analysis of the suggested phenomenon is in general challenging, therefore we do not provide potentially wrong $t(1/2)$ values and show

only “macroscopic” recovery curves of individual proteins similarly as we did before for other coat proteins (Skruzny et al., 2012). We also now use word “low”, not “slow” recovery to do not bias between these two possible scenarios (p.13, par 3).

9. In Fig. 3, the use of red/yellow/green to indicate different proximity networks may be difficult to interpret for individuals who cannot distinguish between these colors.

We thank the reviewer for this point. We tested color schemes of our figures in software simulating the protanopia and deuteranopia color perceptions and changed them accordingly.

10. What is the cutoff value for "no FRET" in this study? On pp.6-7, a range of 1.1-8.9% is implied, but in Table 1, there are some non-zero results that are not reported in Fig. 1. It would be helpful to explicitly define the cutoff value.

We agree with the reviewer and provide now the description of “FRET-positive” cut-off value in Materials and Methods section “Statistics and reproducibility”: Protein pairs were assigned as “FRET positive”, when their mean FRET and both 95% confidence intervals were positive values (see Dataset EV1).

Reviewer #3:

This is an interesting study of the architecture of the clathrin-coated membrane formed in preparation for endocytosis in budding yeast. Technically the work seems to be well done and the authors are careful not to over-interpret their FRET efficiencies as distances. However, some aspects of the interpretation need to be modified, because the authors tend to think in one dimension, while their signals are coming from a 3D network. The interpretative figures also need extensive revision to be more anatomically correct, to take into account stoichiometries and to be more informative.

Page 6: Start new paragraphs with "To estimate.." and "We then analyzed..."

We changed the text as suggested.

Figure 2: State whether the cells are alive or fixed. Provide a better anatomical description of the micrographs. For example, note that most of the spots are clusters of endocytic patches rather than single patches. Please explain why studying such closely packed patches is not a problem. Explain how GFP and mCherry are linked in the tandem construct.

We extended Figure 2 legend accordingly:

...The indicated GFP- and mCherry-tagged endocytic proteins localize at the endocytic patches on the plasma membrane of live yeast cells. The enrichment of individual endocytic patches (diameter of 50-80 nm) on the plasma membrane, caused by the LatA-induced block of their subsequent invagination and/or their

higher abundance at incipient buds, resulted in their apparent clustering on the diffraction-limited fluorescence images (pixel size, 178 nm).

The peptide linkers connecting coat proteins and their fluorescent tags are now detailed in Material and Methods part "Yeast strains and media".

Page 7: Figure 3 is a well-intentioned approach to illustrate the interaction network based on the FRET observations, but this figure is more confusing than edifying. The grey bars are inadequate representations of the protein molecules and not explained. For example, what do the lengths of bars represent? Obviously, it is not the sizes of the proteins, given the way Chc and Clc are depicted. How about taking better advantage of the known structure of clathrin and other proteins to illustrate the findings with scale drawings? For example, a scale drawing of the clathrin molecule (from the coated vesicle atomic model) would show that it extends across the entire figure a known distance from the lipid bilayer. Where other structures or domains are known, they should be depicted in proportion to their sizes and properly oriented relative to the lipid bilayer. For example, the BAR domain of Bzz1 should be horizontal on the bilayer. Consider including the common names of these proteins in the figure, since the gene names can confuse people in the field and are hopeless for outsiders. Consider taking advantage of the UCSF "Modeler" suit of programs to build this figure. Please include directly in the panels the color code for the lines.

We thank the reviewer for these important points and provide now a corrected version of Figure 3. As suggested, we now more properly scale and orient schemes of studied proteins (e.g. clathrin, Bzz1), including, where known, schemes of their domain organization. We also indicate both yeast and human proteins names, and add the color-coding legend as a part of the figure. We hope that the figure now provides sufficient structural and topological detail of depicted proteins without distracting from the key message of detected FRET proximities and their networks. (For this reason and because of an incomplete structural information of studied proteins, we did not aim to show their molecular structures e.g. by Modeller). The current Figure 3 is still coherent with related schemes of functional/regulatory motifs and interactions of the proteins shown in Figure 7, while structurally more correct and better scaled model of the proposed endocytic coat architecture is shown in Figure 6.

Two other issues should be addressed:

(1) How to take into account both the X-Y and Z distances between the FPs? The text on page 9 concludes that "the N-terminus of Sla1 is located more than 10 nm from the clathrin lattice." But, is this 10 nm in Z or in X-Y or some combination of the two. The same problem occurs for each protein pair in the figures and throughout the text. One example of apparent separation in the X-Y plane is the absence of signal between the Bzz1 F-BAR domain and the green network. Again, on page 10, only Z is considered in concluding "a more membrane proximal location of the N-terminus of Pan1..." The same applies to the interpretation of the data in Figure 4.

We now try to better stress our focus on the analysis of the coat architecture in the membrane-cytoplasm direction (Z-axis) in respective parts of Results and Discussion:

(p. 7, par. 2) The FRET proximities detected in our screen allowed us to construct the protein map of the endocytic coat in the plasma membrane-cytoplasm direction (Fig 3A).

(p. 17, par. 2) In addition, the absence of FRET between the coat proteins and proteins localized at the outer rings of the endocytic site (Syp1-Ede1, actin regulators) does not allow us to infer about a potential organization of the coat at the membrane plane axis. Based on the apparent homogeneity of the coat proteins at this plane shown by superresolution studies (Mund et al, 2018), we assume that the coat proteins do not adopt very discrete organization along the flat membrane and the detected FRET proximities predominantly mirror their organization in the plasma-membrane-cytoplasm direction (Fig 6).

In other words, our focus on Z-axis is probably inevitable, as only in this direction we can compare our FRET results with previously established “Z-axis topology” markers (Sla2 termini, clathrin subunits) and make further hypotheses. This is impossible for the axes parallel to the flat membrane (X-Y), where no data about a potential stratification of the coat exist. With known functional architecture of the clathrin lattice and the apparent homogeneity of the yeast coat proteins in the X-Y plane (Mund et al., 2018; <https://doi.org/10.1016/j.cell.2018.06.032>), we find unlikely that our FRET proximities predominantly mirror distances in X-Y directions. If this will be the case, one would also expect some FRET being detected between either Syp1-Ede1 or Bzz1/Las17/Vrp1 outer “rings” and a specific protein of the coat. As we did not obtain such data we currently cannot speculate about the coat organization in the X-Y axes.

(2) The stoichiometry of the components. Figure 3 has one copy of each protein, but Figure 6 has multiple copies. Is either depiction accurate?

Figure 3 focuses on networks of FRET proximities detected between individual coat-associated proteins, therefore each protein is depicted only once to do not draw any connection redundantly. In Figure 6 we tried to discern more and less abundant coat proteins by adding 2-3 or 1-2 copies of them, respectively. Attempts to fill the model with more protein copies, which would better mirror their in vivo stoichiometries, resulted in a very poor discrimination of individual proteins and overall loss of clarity. We now comment about in vivo protein copy number in Figure 6 legend: Note that only few copies of individual coat proteins (presented at the endocytic site in dozens of copies; Picco et al, 2015; preprint: Sun et al, 2019) are shown.

Page 9: Deletion of the C-terminus of fission yeast Pan1 compromises actin patch assembly (Chen, Curr Biol, 2013), so I am skeptical that the Sc Pan1 truncation is fully functional.

According to the Pan1 deletional analysis (Bradford et al., 2015, <https://doi.org/10.1091/mbc.E14-11-1510>) C-terminally deleted Pan1(1-1050) supports the growth and behaves at the endocytic sites as the wild-type Pan1 protein, so its essential function is probably retained (p.8, last sentence). We are aware that this truncation is absent of several identified actin-regulatory motifs (WH2-like, acidic) potentially compromising Pan1 function during actin-dependent steps of endocytosis, but Bradford's data suggest that its function before these steps (where our proximity mapping was focused) is not substantially compromised. To study Pan1 rearrangements during actin-dependent invagination we then specifically used a shorter Pan1 truncation Pan1(1-1303), which kept all actin-regulatory motifs except poly-proline stretch potentially interacting with Myo3/5 (Barker et al., 2007; <https://doi.org/10.1091/mbc.e07-05-0436>). We cannot not use the full-length Pan1 in these studies, as no FRET partner was detected for its probably too remote, intact C-terminus.

Page 11: What is the meaning of "previously described cytoplasmic interactions of coat-associated proteins?" Is this FRET signal coming from the dispersed phase of the cytoplasm or from the cytoplasm adjacent to a site of endocytosis?

We corrected the sentence: Indeed, we could detect cytoplasmic interactions of coat-associated proteins (Syp1-Ede1, Pan1-End3, and Las17-Sla1) previously found by FCCS (Boeke et al., 2014)...

We now specify in Materials and Methods section "FRET Microscopy" that we used:
 "...inner cytoplasmic (nucleus and vacuole free) areas for studies of cytoplasmic FRET."

Figure 4: The relative scales of the membrane invagination, clathrin coat, actin network and the proteins are incorrect by more than an order of magnitude, so the depiction is misleading.

We corrected the scales of the clathrin lattice and the actin network on the scheme to be more realistic.

The discussion is simply a restatement of the data and interpretations already presented in the results section. This redundancy is not necessary. Perhaps all of the interpretation can be moved to the discussion.

We wish to disagree with this general statement. We found our short conclusive sentences in Results sections essential for a logical build up of the subsequent experiments. In the Discussion, these conclusions are extensively interpreted, compared with the literature and finally used to build the comprehensive functional model of the endocytic coat.

Thank you for sending us your revised manuscript. We have now heard back from two of the three referees who agreed to evaluate your manuscript. Unfortunately, after a series of reminders we did not manage to obtain a report from reviewer #1. In the interest of time, and since the recommendations of referee #2 and #3 are quite similar, I prefer to make a decision now rather than further delaying the process.

[NOTE: MSB did receive the referee report from reviewer #1 on the 1st of November 2019 and has been inserted into this Editorial decision]

You will see from the comments below that reviewer #2 and #3 think that while the majority of the concerns raised by all three reviewers have been addressed, several important issues remain. In principle, our editorial policy only allows a single round of major revision. However, we think it is important to experimentally address reviewer #2's concern with regard to the direct interaction between the tagged proteins and clathrin. We would therefore ask you to address this point together with other comments from both reviewers in an exceptional second round of revision.

REFEREE REPORTS

Reviewer #1

Skruzny and co-authors addressed most of my comments, clarified the text and provided missing information, that in general improved the manuscript clarity.

Unfortunately, the authors confirmed that the experiments were repeated only twice and statistics calculated from the pooled data. As a general principle, I find this is below the gold standards of good scientific practice. Specifically, the number of patches in the pooled set was in the range 40-100. Such approach is known to be prone to underestimation uncertainty, as I pointed out in the initial report. Although some results, such as those in Fig.4C, will probably stand under a more rigorous analysis, I am not convinced that others can yield conclusive evidence. For example, a proper analysis (each repeat analyzed independently and then the result merged) will have to confirm the significance of the statements based on Fig.4B ("Partial loss of FRET between Ent1-mTurquoise2 and Sla1-mNeonGreen").

The "raw" data in suppl. table Dataset EV1 raise questions: how the authors get confidence interval in FRET signals smaller than uncertainties in correction for direct donor bleaching? For example: GFP-YAP1801 has uncertainty of correction of direct donor bleaching $\pm 1.7\%$ and uncertainty of FRET with acceptor Ent1-mCherry $\pm 0.8\%$, etc.

In summary, I remain concerned with the conclusions drawn for the data and am not persuaded that this study fulfills the quality standard of a journal on Systems Biology like MSB.

Reviewer #2:

In the present manuscript, Skruzny and colleagues use a FRET approach to assess proximity between pairs of proteins involved in clathrin-mediated endocytosis in yeast. This approach provides valuable information on the relative position of individual proteins and their N- and C-termini compared to other proteins as well as the plasma membrane. Many prior studies have provided convincing evidence favoring modular assembly of CME machinery in order to concentrate cargos prior to CCP budding and generation of an endocytic vesicle. However, the precise architecture of the endocytic machinery has not been well-elucidated. The data presented here may provide some

insight into how CME proteins act in concert.

The present revision of the manuscript has provided a number of important clarifications and additional data that strengthen the study as a whole. In general, the majority of comments and concerns raised by three reviewers have been addressed; however, there remain some issues based on the initial reviews that I believe need clarification and/or additional experimentation:

1. The authors have now provided information about the linker sequence between the C-termini of Ent1/2 and Yap1801/2 and the N-terminal portion of their fluorescent tags, with specification that an acidic residue was indeed included in place of the stop codon to preserve the clathrin-binding motif. However, the authors chose not to verify that the tagged proteins can still directly interact with clathrin. This was an important point, because it remains possible that a bulky fluorescent protein tag could occlude the clathrin-binding motif, even where an acidic amino acid and a linker sequence is present. For assessing proximity by FRET, it is important to know that these tagged proteins behave essentially the same way that their untagged counterparts do. This is especially true where there is a real possibility that a relevant binding motif is no longer functional, and recruitment is instead mediated by interaction with other proteins in the CME network. In other words, if the C-terminus isn't functional, proximity data may or may not be accurate and reflective of the protein's true behavior.

Recombinant epsins have been purified from *E. coli* (Aguilar 2003, *J Biol Chem* 278:10737-43) using standard methods and remain functional in their ability to bind other proteins. Given that any proximity data for the adaptors is based on the assumption that the proteins bind to clathrin, verifying this interaction directly for the tagged proteins for at least one of the adaptors is an important step toward validating that they retain function at their C-termini.

2. The authors' reply to reviewer 1 (page 1 of their response) raises some concerns about how the data were analyzed. Specifically, they report "FRAP and FRET studies of protein rearrangements were performed 3 times with 3+ patches/cells and 22+ patches (8+ cells) per session, respectively." The numbers for FRET analysis (22+ patches over 8+ cells per session) would suggest an average of 2.75 patches analyzed per cell; however, the large number of prior studies on yeast endocytosis, as well as the data in Fig. 2 here, convincingly show that there are many more endocytic sites at any time in a single cell. How have the authors chosen which patches to analyze? For measurements to be unbiased, all observable individual patches should be included in the analysis for each cell being analyzed. If this was the case, it should be stated more clearly.

This may have been addressed in the authors' response to reviewer 1's final point (page 3 of response letter), but it still is not immediately clear if all patches within each cell analyzed were included (and the number of patches per cell still is far lower than expected).

Reviewer #3:

The authors responded constructively to most of my concerns. However, a few remain:

(1) How does one account distances between the FPs in both the X-Y plane and Z direction perpendicular to the membrane?

The authors clarified that their data only provides information in the Z direction, but this important point might be missed in the following: "The FRET proximities detected in our screen allowed us to construct the protein map of the endocytic coat in the plasma membrane-cytoplasm direction (Fig 3A)."

I suggest the following wording "We used the FRET proximities from our screen to construct an interaction map of the proteins of the endocytic coat in the direction normal to the plasma membrane (Fig 3A). Our data do not provide information about interactions in the plane parallel to the plasma membrane."

On page 17 and throughout, consider replacing "plasma-membrane-cytoplasm direction" with "direction normal to the plasma membrane" or "direction perpendicular to the plasma membrane."

Similarly, consider replacing throughout "the membrane plane axis" with "in the plane of the membrane" or "in planes parallel to the membrane."

(2) The stoichiometry of the components.

The legend should state that Figure 3 shows only one copy of each protein that actually present in multiple copies. I might be worth noting the stoichiometries in the text or legend.

2nd Revision - authors' response

14th February 2020

Response to reviewers

Reviewer #1

Skruzny and co-authors addressed most of my comments, clarified the text and provided missing information, that in general improved the manuscript clarity.

Unfortunately, the authors confirmed that the experiments were repeated only twice and statistics calculated from the pooled data. As a general principle, I find this is below the gold standards of good scientific practice. Specifically, the number of patches in the pooled set was in the range 40-100. Such approach is known to be prone to underestimation uncertainty, as I pointed out in the initial report. Although some results, such as those in Fig.4C, will probably stand under a more rigorous analysis, I am not convinced that others can yield conclusive evidence. For example, a proper analysis (each repeat analyzed independently and then the result merged) will have to confirm the significance of the statements based on Fig.4B ("Partial loss of FRET between Ent1-mTurquoise2 and Sla1-mNeonGreen").

We initially performed only two rounds of the FRET screen due to its huge time requirements and because of similar results obtained in both rounds. It may not be apparent from the text, but imaging and analysis of each strain requires approx. 2-3 hours of work, which with 217 screened strains represents a serious work load for each new repetition. All other experiments were then performed in three independent replicates.

We understand a potential concern about the statistical robustness of our screen and performed a third round of the screen with the majority of studied protein pairs. Specifically, we rescreened 122 strains covering all possible proximities between endocytic coat proteins, as well as all potential proximities between coat proteins and Ede1/Syp1 or actin regulators Las17, Lsb3, Bzz1 and Vrp1 suggested from literature (gray lines in Fig. 5). We did not rescreen strains of Ede1/Syp1 proteins with clearly low colocalization with other proteins (35 strains), and protein pairs of actin regulators showing "no FRET" signal in the two initial rounds (60 strains) as these were not further followed in our work (discussed on p. 17).

The third screen increased the number of analyzed patches to 80-200 (from 10-40 cells). Most importantly, the new data did not change any outcome of the original dataset. FRET% numbers were only modestly changed (see Fig. 1) and no additional FRET proximities were detected or lost.

Apart from comparable results of each round of the screen, many other lines of evidence suggest its robustness: i) Almost all termini of endocytic coat proteins have clear “node” character constituting well closed network of FRET proximities inside a particular proximity layer (Fig. 3A). ii) The identical set of proximities was obtained for differentially expressed, but functionally similar Ent1/Ent2 and Yap1801/2 protein duplicates (p. 21, Dataset EV1). iii) Similar FRET results were detected for all protein pairs with interchanged fluorophores (p. 21).

Pooling of similar datasets is probably inevitable for fluorescence screens of this extend, especially when they are focused on small objects providing only limited number of fluorescence counts (in our case further narrowed by the FRET readout and by the small photobleached area covering just 2-5 cells). This procedure have been succesfully applied in many landmark papers of the field using comparable patch number for their calculations (e.g. Kaksonen et al., 2005; Picco et al., 2015; Sun et al., 2020).

To further support the results obtained in Fig. 4B, where the detection of significant FRET change between Ent1 and Sla1 proteins at early vs. late endocytic sites required analysis of all 2x74 patches, we performed a new round of this experiment. The detected FRET difference was again very similar to the first dataset ($p=0.037$ vs. $p=0.028$). We provide the new dataset as Appendix Fig. 1. We also wish to stress our critical view on this small FRET change in Discussion (p. 17).

All details of FRET screening with corrected cell/patches numbers are now detailed in Materials and Methods on p. 23. All raw data with individual replicates indicated are provided as Source data. All raw images will be available on request.

The "raw" data in suppl. table Dataset EV1 raise questions: how the authors get confidence interval in FRET signals smaller than uncertainties in correction for direct donor bleaching? For example: GFP-YAP1801 has uncertainty of correction of direct donor bleaching $\pm 1.7\%$ and uncertainty of FRET with acceptor Ent1-mCherry $\pm 0.8\%$, etc.

This phenomenon is typical for few lowly expressed donor proteins with not easily discernible localization at the endocytic sites (Yap1801 and Apl1). These proteins have either more diffuse membrane signal (either naturally or due to their low fluorescence signal-to-noise ratio) or localize to other membrane structures. In donor-only strains (absent of any additional endocytic marker), the protein signal of endocytic sites cannot be always clearly discerned from the signal of other membrane sites, and these could be sometimes mistakenly taken for FRET calculations. In addition, in case of FRET, fluorescence of these donors increases after photobleaching, reducing thus their non-favorable signal-to-noise ratio specifically in case of proximal acceptor (e.g. Ent1).

Reviewer #2

1. The authors have now provided information about the linker sequence between the C-termini of Ent1/2 and Yap1801/2 and the N-terminal portion of their fluorescent tags, with specification that an acidic residue was indeed included in

place of the stop codon to preserve the clathrin-binding motif. However, the authors chose not to verify that the tagged proteins can still directly interact with clathrin. This was an important point, because it remains possible that a bulky fluorescent protein tag could occlude the clathrin-binding motif, even where an acidic amino acid and a linker sequence is present. For assessing proximity by FRET, it is important to know that these tagged proteins behave essentially the same way that their untagged counterparts do. This is especially true where there is a real possibility that a relevant binding motif is no longer functional, and recruitment is instead mediated by interaction with other proteins in the CME network. In other words, if the C-terminus isn't functional, proximity data may or may not be accurate and reflective of the protein's true behavior.

Recombinant epsins have been purified from *E. coli* (Aguilar 2003, *J Biol Chem* 278:10737-43) using standard methods and remain functional in their ability to bind other proteins. Given that any proximity data for the adaptors is based on the assumption that the proteins bind to clathrin, verifying this interaction directly for the tagged proteins for at least one of the adaptors is an important step toward validating that they retain function at their C-termini.

After discussion with the editor, we tested the interaction between the FP-tagged adaptor proteins and clathrin in vivo. Specifically, we employed previously published two-hybrid interaction between Ent1/Ent2 and the N-terminal domain of clathrin heavy chain (Chc1-TD), which was shown to depend on adaptor's C-terminal clathrin-binding motif by S. Lemmon lab (Collette et al., 2009). We obtained Ent1-/Ent2-GFP fusions directly from the genome of screened strains and cloned them into respective two-hybrid vectors. As now shown in Appendix Fig. S2 both GFP fusions interacted with Chc1-TD similarly to non-tagged proteins.

2. The authors' reply to reviewer 1 (page 1 of their response) raises some concerns about how the data were analyzed. Specifically, they report "FRAP and FRET studies of protein rearrangements were performed 3 times with 3+ patches/cells and 22+ patches (8+ cells) per session, respectively." The numbers for FRET analysis (22+ patches over 8+ cells per session) would suggest an average of 2.75 patches analyzed per cell; however, the large number of prior studies on yeast endocytosis, as well as the data in Fig. 2 here, convincingly show that there are many more endocytic sites at any time in a single cell. How have the authors chosen which patches to analyze? For measurements to be unbiased, all observable individual patches should be included in the analysis for each cell being analyzed. If this was the case, it should be stated more clearly.

This may have been addressed in the authors' response to reviewer 1's final point (page 3 of response letter), but it still is not immediately clear if all patches within each cell analyzed were included (and the number of patches per cell still is far lower than expected).

We apologize for this misunderstanding. We used cca 22 Abp1-positive and 22 Abp1-negative patches of 8 cells, so around five patches per cell in total. The only prerequisite for patch selection was its isolated position on the cell cortex to avoid the interference of its fluorescence with the fluorescence of adjacent patches. This restriction fullfiled on avarage 2-3 Abp1-positive patches per cell, the same number

of more abundant Abp-negative patches was then randomly taken from the same cell. We clarified this in Material and Methods section, p. 23.

Reviewer #3:

The authors responded constructively to most of my concerns. However, a few remain:

(1) How does one account distances between the FPs in both the X-Y plane and Z direction perpendicular to the membrane?

The authors clarified that their data only provides information in the Z direction, but this important point might be missed in the following: "The FRET proximities detected in our screen allowed us to construct the protein map of the endocytic coat in the plasma membrane-cytoplasm direction (Fig 3A)."

I suggest the following wording "We used the FRET proximities from our screen to construct an interaction map of the proteins of the endocytic coat in the direction normal to the plasma membrane (Fig 3A). Our data do not provide information about interactions in the plane parallel to the plasma membrane."

On page 17 and throughout, consider replacing "plasma-membrane-cytoplasm direction" with "direction normal to the plasma membrane" or "direction perpendicular to the plasma membrane." Similarly, consider replacing throughout "the membrane plane axis" with "in the plane of the membrane" or "in planes parallel to the membrane."

We thank the reviewer for suggested wording ideas. We corrected the text accordingly.

(2) The stoichiometry of the components.

The legend should state that Figure 3 shows only one copy of each protein that actually present in multiple copies. I might be worth noting the stoichiometries in the text or legend.

We agree with this suggestion. As our work does not directly determine stoichiometries of analyzed proteins, we refer to two exhaustive studies on this topic (Picco et al., 2015; Sun et al., 2020).

3rd Editorial Decision

19th March 2020

Thank you for sending us your revised manuscript. We have now heard back from the two reviewers who were asked to evaluate your study. As you will see the reviewers are satisfied with the modifications made and think that the study is now suitable for publication.

Before we formally accept your manuscript, we would ask you to address a few remaining editorial issues listed below.

REFeree REPORTS

Reviewer #1:

The authors have revised the study. Previously, they had not enough repeats of the experiments and I questioned how reliable were the results. The authors have repeated the experiments for the hits and argued that repeat for the negative is too time consuming. The new data (3d repeat) confirmed the data of previous revision of the manuscript. So, the response to my previous criticism is formally met and the ms can be accepted for publication.

Reviewer #3:

I am satisfied with the revisions. The review process has greatly strengthened this paper.

3rd Revision - authors' response

2nd April

The Authors have made the requested editorial changes.

Accepted

3rd April

Thank you again for sending us your revised manuscript. We are now satisfied with the modifications made and I am pleased to inform you that your paper has been accepted for publication.

Corresponding Author Name: Skruzny Michal

Manuscript Number: MSB-19-9009RR